# HyperMask: Adaptive Hypernetwork-based Masks for Continual Learning

## Abstract

Artificial neural networks suffer from catastrophic forgetting when they are sequentially trained on multiple tasks. To overcome this problem, there exist many continual learning strategies. One of the most effective is the hypernetwork-based approach. The hypernetwork generates the weights of a target model based on the task's identity. The model's main limitation is that hypernetwork can produce completely different nests for each task. Consequently, each task is solved separately. The model does not use information from the network dedicated to previous tasks and practically produces new architectures when it learns the subsequent tasks. To solve such a problem, we use the lottery ticket hypothesis, which postulates the existence of sparse subnetworks, named winning tickets, that preserve the performance of a full network.

In the paper, we propose a method called HyperMask, which trains a single network for all tasks. Hypernetwork produces semi-binary masks to obtain target subnetworks dedicated to new tasks. This solution inherits the ability of the hypernetwork to adapt to new tasks with minimal forgetting. Moreover, due to the lottery ticket hypothesis, we can use a single network with weighted subnets dedicated to each task.

## 1 Introduction

Learning from a continuous data stream is challenging for deep learning models. Artificial neural networks suffer from catastrophic forgetting (McCloskey & Cohen, 1989) and drastically forget previously known information upon learning new knowledge. Continual learning (CL) Hsu et al. (2018) effectively learns consecutive tasks, preventing forgetting already learned ones. Continuous learning is a rapidly developing field of machine learning that utilizes various techniques.

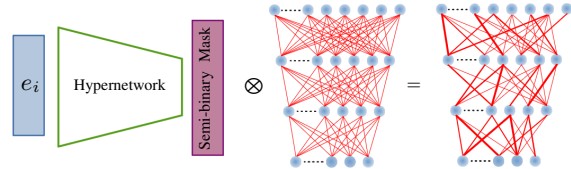

Figure 1: Commonly, the parameters of a neural network are directly adjusted from data to solve a task. In HyperMask hypernetwork maps embedding vectors $e_i$ to semi-binary mask, which produces a subnetwork dedicated to the target network to solve the $i$–th task.

Regularization-based methods (Kirkpatrick et al., 2017; Chaudhry et al., 2020; Jung et al., 2020; Titsias et al., 2019; Mirzadeh et al., 2020) aim to keep the learned information about previous tasks by regularizing it to previous weights. Rehearsal-based methods Rebuffi et al. (2017); Chaudhry et al. (2018); Saha et al. (2020) use a set of real or generated data from previous tasks. Architecture-based approaches Mallya et al. (2018); Serra et al. (2018); Li et al. (2019); Wortsman et al. (2020); Kang et al. (2022) suggest that interference between tasks can be reduced by using newly developed architectural elements.

The hypernetwork von Oswald et al. (2019); Henning et al. (2021) approach is located at the crossroads of regularization-based and architecture-based approaches. A hypernetwork architecture Ha et al. (2016) is a neural network that generates weights for a separate target network designated to solve a specific task. In a continual learning setting, a hypernetwork generates the weights of a target model based on the task identity. Such models can be considered an architecture-based

approach, since we build a new architecture for each task. On the other hand, we can treat hypernetwork like a regularization model. At the end of training, we have a single meta-model, which produces dedicated weights. Due to the ability to generate completely different weights for each task, hypernetwork-based models feature minimal forgetting. Unfortunately, such properties were obtained by producing completely different architectures for substantial tasks. Only hypernetworks uses information on tasks. Such a model can produce different nests for each task and solve them separately. The hypernetwork cannot use the weight of the target network from the previous task.

To solve such a problem, we use the lottery ticket hypothesis (LTH) Frankle & Carbin (2018), which postulates that we can find subnetworks named winning tickets with performance similar (or even better) to the full architecture. However, the search for optimal winning tickets in continual learning scenarios is difficult Mallya et al. (2018); Wortsman et al. (2020), as iterative pruning requires repetitive pruning and retraining for each arriving task, which is impractical. Alternatively, Winning SubNetworks (WSN) Kang et al. (2022) incrementally learns model weights and task-adaptive binary masks. WSN eliminates catastrophic forgetting by freezing the subnetwork weights considered important for the previous tasks and memorizing masks for all tasks.

Our paper proposes a method called HyperMask[1] which combines hypernetwork and lottery ticket hypothesis paradigms. Hypernetwork produces semi-binary masks to the target network to obtain weighted subnetworks dedicated to new tasks; see Fig. 1. The masks produced by the hypernetwork modulate the weights of the main network and act like dynamic filters, enhancing the target weights that are important for a given task and decreasing the importance of the remaining weights. In consequence, we work on a single network with subnetworks dedicated to each task and we do not need to freeze any part of this model. When HyperMask learns a new task, we reuse the learned subnetwork weights from the previous tasks. HyperMask also inherits the ability of the hypernetwork to adapt to new tasks with minimal forgetting. We produce a semi-binary mask directly from the trained task embedding vector, which creates a dedicated subnetwork for each dataset.

To the best of our knowledge, our model is the first architecture-based CL model that uses hypernetwork, or, in general, any meta-model, for producing masks for other networks. Updates of hypernetworks are prepared not directly for the weights of the main model, like in von Oswald et al. (2019), but for masks dynamically filtering the target model.

Our contributions can be summarized as follows:

- We propose a method that uses the hypernetwork paradigms for modeling the lottery ticket-based subnetwork. The hypernetwork modulates the weights of the main model instead of their direct preparation as in von Oswald et al. (2019).
- HyperMask inherit the ability to reuse weights from the lottery ticket module and adapt to new tasks from the hypernetwork paradigm.
- The semi-binary mask of HyperMask helps the target network to discriminate classes in consecutive CL tasks, see Fig. 3.

## 2 RELATED WORKS

**Continual learning** Typically, continual learning approaches are divided into three main categories: regularization, dynamic architectures, and replay-based techniques (Parisi et al., 2019; De Lange et al., 2021; Wang et al., 2023).

Regularization-based techniques expand the loss function by using regularization terms that control the distance between optimal parameters from the previous task and the new one. We hypothesize that the best parameters for a new task can be located in the neighborhood of nest parameters from prior tasks. In the case of weight regularization, we regularize the variation of the most important network parameters. In EWC (Kirkpatrick et al., 2017; Ritter et al., 2018), the importance is expressed by the Fisher information matrix. SI (Zenke et al., 2017) approximates the contribution of the parameter to the total loss variation and its update length throughout the training trajectory. MAS (Aljundi et al., 2018) accumulates importance measurements based on the sensitivity of predictive results to changes in parameters, both online and unsupervised.

---

[1]The source code is available at `https://github.com/...`

In the case of function regularization, we use the regularization term not on weights but on the intermediate or final output of the prediction function. In the learning without forgetting paradigm (LwF) (Li & Hoiem, 2017), we use distillation loss to compare new task outputs generated by the new and old models. LwM (Dhar et al., 2019) takes advantage of attention maps for training samples. EBLL (Jung et al., 2020) learns task-specific autoencoders and prevents changes in feature reconstruction. In CW-TaLaR (Mazur et al., 2022), we use the Cramer-Wold distance Knop et al. (2020) between two probability distributions defined in a target layer of an underlying neural network shared by all tasks.

Rehearsal-based approaches store information about data for training previous tasks and replay them to prevent catastrophic forgetting. In experience replay, we typically store a few old training samples within a small memory buffer. Reservoir Sampling (Riemer et al., 2018; Chaudhry et al., 2019) randomly selects a fixed number of old training samples obtained from each training batch. A Ring Buffer (Lopez-Paz & Ranzato, 2017) guarantees that the same amount of old training samples is present for each class. Mean-of-Feature (Rebuffi et al., 2017) selects a similar number of old training samples that are closest to the mean of the features of each class. In generative replay or pseudo-rehearsal, we train an additional generative model to replay generated data. DGR (Shin et al., 2017) provides an initial framework for data sampling from the old generative model to inherit previously learned knowledge. MeRGAN (Wu et al., 2018) enforces the consistency of the generated data with the same random noise between the old and new generative models, similar to the role of function regularization.

Architecture-based approaches use dynamic architectures that dedicate separate model branches to different tasks. These branches can be developed incrementally, such as in the case of Progressive Neural Networks Rusu et al. (2016). The architecture of a system can be optimized to enhance parameter effectiveness and knowledge transfer, for example, by reinforcement learning (RCL (Xu & Zhu, 2018), BNS (Qin et al., 2021)), architecture search (LtG (Li et al., 2019), BNS (Qin et al., 2021)), and variational Bayesian methods (BSA (Kumar et al., 2021)). Alternatively, a static architecture can be reused with iterative pruning as proposed by PackNet (Mallya & Lazebnik, 2018) or by the application of Supermasks (Wortsman et al., 2020).

**Pruning-based Continual Learning**  Most architecture-based methods use additional memory to obtain better performance of continual learners. In the pruning-based method, we build computationally efficient and memory-efficient strategies.

CLNP (Golkar et al., 2019) freezes the most significant neurons for a given task. Then, we reinitialize weights that were not selected for future task training. Piggyback (Mallya et al., 2018) uses a pre-trained model and task-specific binary masks. This technique has limited knowledge transfer since we retrain the binary masks for each task. Consequently, the approach's effectiveness largely depends on the caliber of the backbone model. HAT (Serra et al., 2018) uses task-specific learnable attention vectors to recognize significant weights for each task.

LL-Tickets (Chen et al., 2020) show that we can find a subnetwork, referred to as lifelong tickets, that performs well on all tasks during continual learning. If the tickets cannot work on the new task, the method looks for more prominent tickets from the existing ones. However, the LL-Tickets expansion process is made up of a series of retraining and pruning steps.

In Winning SubNetworks (WSN) Kang et al. (2022), authors propose to jointly learn the model and task-adaptive binary masks dedicated to task-specific subnetworks (winning tickets). Unfortunately, WSN eliminates catastrophic forgetting by freezing the subnetwork weights for the previous tasks and memorizing masks for all tasks.

This paper proposes the next step toward producing a sparse subnetwork for continual learning. Instead of the classical binary mask and freezing strategy, we use the hypernetwork paradigm. The hypernetwork generates a semi-binary mask to a target model based on the task embedding.

**Hypernetworks for continual learning**  A hypernetwork architecture Ha et al. (2016) is a neural network that generates a vector of weights for a separate target network designated to solve a specific task. Hypernetworks are widely used, , *e.g.*, generative models Spurek et al. (2020), implicit representation Szatkowski et al. (2023) and few-shot learning Sendera et al. (2023).

In a continuous learning environment, a hypernetwork generates the weights of a target model based on the task's identity. HNET von Oswald et al. (2019) uses task embeddings to produce weights dedicated to each task. HNET can be seen as an architecture-based strategy as we create a distinct architecture for each task, but it can also be viewed as a regularization model. After training, a single meta-model is left, which produces specialized weights. Thanks to the possibility of generating completely different weights for each task, hypernetwork-based models demonstrate minimal forgetting. However, this advantage leads to difficulty with forward/backward transfers. Hypernetworks can generate different nests for tasks and solve them independently. Consequently, the hypernetwork may have problems using the previously learned knowledge to solve a new task. In Henning et al. (2021), authors propose a Bayesian version of the hypernetworks in which they produce parameters of the prior distribution of the Bayesian network.

## 3 HYPERMASK: ADAPTIVE HYPERNETWORKS FOR CONTINUAL LEARNING

This section describes our hypernetwork-based continual learning method called HyperMask. In HyperMask, the hypernetwork returns semi-binary masks to produce weighted subnetworks dedicated to new tasks. This solution inherits the ability of the hypernetwork to adapt to new tasks with minimal forgetting. Moreover, we can use a single network with weighted subnets dedicated to each task thanks to the lottery ticket hypothesis.

**Problem statement** Let us consider a supervised learning setup where $T$ tasks are derived to a learner sequentially. We denote that $X_t = \{x_{i,t}\}_{i=1}^{n_t}$ is the dataset for the task $t$, composed of $n_t$ elements of raw instances and $Y_t = \{y_{i,t}\}_{i=1}^{n_t}$ are the corresponding labels. Data from all tasks we denote by $D_t = (X_t, Y_t) \subset X \times Y$. We assume a neural network $f(\cdot; \boldsymbol{\theta})$, parameterized by the model weights $\boldsymbol{\theta}$ and the standard continual learning scenario

$$\boldsymbol{\theta}^* = \text{minimize}_{\boldsymbol{\theta}} \frac{1}{n_t} \sum_{i=1}^{n_t} \mathcal{L}\left(f(\boldsymbol{x}_{i,t}; \boldsymbol{\theta})\right),$$

where $\mathcal{L}(\cdot, \cdot)$ is a classification objective loss such as the cross-entropy loss. $D_t$ for task $t$ is only accessible when learning task $t$, but repetition-based continual learning methods allow memorization of a small portion of the dataset to replay. We further assume that task identity is given in both the training and testing stages, except for the additional series of experiments.

To provide space for learning future tasks, a continuing learner often adopts over-parameterized deep neural networks. We can find subnets with equal or better performance assuming overly parametric depth neural networks. In our model, we use a hypernetwork paradigm to produce subnets.

**Hypernetwork** Hypernetworks, introduced in Ha et al. (2016), are defined as neural models that generate weights for a separate target network solving a specific task. Before we present our solution, we describe the classical approach to using hypernetworks in CL. A hypernetwork generates individual weights for all tasks in a continual learning setting. In HNET von Oswald et al. (2019); Henning et al. (2021) the authors propose using trainable embeddings $\boldsymbol{e}_t \in \mathbb{R}^N$, for $t \in \{1, ..., T\}$, and the hypernetwork $\mathcal{H}$ with weights $\boldsymbol{\Phi}$ generating weights $\boldsymbol{\theta}_t$ for the target network $f$ dedicated to the $t$–th task

$$\mathcal{H}(\boldsymbol{e}_t; \boldsymbol{\Phi}) = \boldsymbol{\theta}_t.$$

HNET meta-architecture (hypernetwork) produces different weights for each continual learning task. We have the function $f_{\boldsymbol{\theta}_t} : X \to Y$ (a neural network classifier with weights $\boldsymbol{\theta}_t$), which takes elements from a continuous learning dataset and predicts labels.

The target network is not trained directly. In HNET, we use a hypernetwork $H_{\boldsymbol{\Phi}} : \mathbb{R}^N \ni \boldsymbol{e}_t \to \boldsymbol{\theta}_t$, which for a task embedding $\boldsymbol{e}_t$ returns weights $\boldsymbol{\theta}_t$ to the corresponding target network $f_{\boldsymbol{\theta}_t} : X \to Y$. Thus, each continual learning task is represented by a function (classifier)

$$f(\cdot; \boldsymbol{\theta}_t) = f(\cdot; H(\boldsymbol{e}_t; \boldsymbol{\Phi})).$$

At the end of training, we have a single meta-model, which produces dedicated weights. Due to the ability to generate completely different weights for each task, hypernetwork-based models feature

minimal forgetting. Hypernetworks can produce different nests for each task and solve them separately. We practically produce a new architecture when we update the prior task. To solve such a problem, we use the lottery ticket hypothesis, which postulates the existence of sparse subnetworks, named winning tickets, that preserve the performance of a full network.

---

**Algorithm 1:** The pseudocode of HyperMask.

---

**Input:** hypernetwork $\mathcal{H}$ with weights $\boldsymbol{\Phi}$,
target network $f$ with weights $\boldsymbol{\theta}$,
sparsity $p \geq 0$, regularization strength
$\beta > 0$, and $\lambda > 0$, $n$ training
iterations, datasets $\{D_1, D_2, ..., D_T\}$,
$(\boldsymbol{x}_{i,t}, y_{i,t}) \in D_t, t \in \{1, ..., T\}$
**Output:** updated hypernetwork weights $\boldsymbol{\Phi}$,
updated target network weights $\boldsymbol{\theta}$

1   Initialize randomly weights $\boldsymbol{\Phi}$ and $\boldsymbol{\theta}$ with embeddings $\{\boldsymbol{e}_1, \boldsymbol{e}_2, ..., \boldsymbol{e}_T\}$;
2   **for** $t \leftarrow 1$ **to** $T$ **do**
3    **if** $t > 1$ **then**
4     $\boldsymbol{\theta}^* \leftarrow \boldsymbol{\theta}$;
5     **for** $t' \leftarrow 1$ **to** $t - 1$ **do**
6      Store $\boldsymbol{m}_{t'} \leftarrow \mathcal{H}(\boldsymbol{e}_{t'}, p; \boldsymbol{\Phi})$;
7     **end**
8    **end**
9    **for** $i \leftarrow 1$ **to** $n$ **do**
10     $\boldsymbol{m}_t \leftarrow \mathcal{H}(\boldsymbol{e}_t, p; \boldsymbol{\Phi})$;
11     $\boldsymbol{\theta}_t \leftarrow \boldsymbol{m}_t \odot \boldsymbol{\theta}$;
12     $\hat{y}_{i,t} \leftarrow f(\boldsymbol{x}_{i,t}; \boldsymbol{\theta}_t)$;
13     **if** $t = 1$ **then**
14      $\mathcal{L} \leftarrow \mathcal{L}_{current}$;
15     **end**
16     **else**
17      $\mathcal{L} \leftarrow \mathcal{L}_{current} + \beta \cdot \mathcal{L}_{output} + \lambda \cdot \mathcal{L}_{target}$;
18     **end**
19     Update $\boldsymbol{\Phi}$ and $\boldsymbol{\theta}$;
20    **end**
21    Store $\boldsymbol{e}_t$;
22   **end**

---

**HyperMask – overview**    Now we are ready to present HyperMask. Our approach uses hypernetwork to produce semi-binary masks for the target network.

We use the *tanh* activation function on the output of Hypernetwork. Then, we select the $p\%$ weights with the highest weight scores, where $p$ is the ratio of target layer capacity and $c(p, i, t; \boldsymbol{x})$ is a threshold value for the $i$-th iteration of the $t$-th task for a given network layer $\boldsymbol{x}$ and $t \in \{1, ..., T\}$. The selection of weights are represented by a task-dependent semi-binary weight mask $\boldsymbol{m}_t$, where an absolute value greater than the threshold denotes that the weight is taken into account during the forward pass and zero otherwise. Formally, $\boldsymbol{m}_t$ is obtained by applying an indicator function $\sigma_p(\cdot; \cdot)$ to a weight $w$ which is an element of $\boldsymbol{x}$ representing a single layer of the hypernetwork $\mathcal{H}$ output

$$\sigma_p(w; \boldsymbol{x}) = \begin{cases} 0 & \text{if } |w| \leq c(p, i, t; \boldsymbol{x}), \\ w & \text{otherwise.} \end{cases}$$

Additionally, the ratio $p$ is constant starting from the second task but, for the first trained task, is gradually increased from 0 to $p$

$$c(p, i, t; \boldsymbol{x}) = \begin{cases} P(p; |\boldsymbol{x}|) & \text{if } t > 1, \\ P(\frac{i}{n} \cdot p; |\boldsymbol{x}|) & \text{otherwise.} \end{cases}$$

Each task is trained through $n$ iterations. The absolute value of consecutive weights of $\boldsymbol{x}$ is calculated element-wise. $P(p; |\boldsymbol{x}|)$ represents the $p$-th percentile of the set of absolute values of a given mask layer.

HyperMask uses trainable embeddings $\boldsymbol{e}_t \in \mathbb{R}^N$ for $t \in \{1, ..., T\}$, threshold level $p$ and hypernetwork $\mathcal{H}$ with weights $\boldsymbol{\Phi}$ generating a semi-binary mask $\boldsymbol{m}_t$ with $p\%$ zeros for the target network weights $\boldsymbol{\theta}$ dedicated to each task

$$\mathcal{H}(\boldsymbol{e}_t, p; \boldsymbol{\Phi}) = \sigma_p(:, \mathcal{H}(\boldsymbol{e}_t; \boldsymbol{\Phi})) = \boldsymbol{m}_t;$$

$\sigma_p(:, \cdot)$ means that the indicator function is applied for all values at the output of $\mathcal{H}$.

In HyperMask, we have two trainable architectures. Hypernetwork $\mathcal{H}$ has trainable parameters $\boldsymbol{\Phi}$, and the target network has trainable parameters $\boldsymbol{\theta}$. Meta-architecture (hypernetwork) produces different semi-binary masks for each continual learning task.

More precisely, we model the function $f_{\boldsymbol{\theta}} : X \to Y$ with general weights $\boldsymbol{\theta}$ dedicated to all tasks. The target network is trained with a classical cross-entropy cost function. We simultaneously train a hypernetwork $H_{\boldsymbol{\Phi}} : \mathbb{R}^N \ni \boldsymbol{e}_t \to \boldsymbol{m}_t$, which for a task embedding $\boldsymbol{e}_t$ returns semi-binary mask $\boldsymbol{m}_t$ to the corresponding target network weights $\boldsymbol{\theta}$. Thus, each continual learning task is represented by a function (classifier)

$$f(\cdot; \boldsymbol{\theta} \odot \boldsymbol{m}_t) = f(\cdot; \boldsymbol{\theta} \odot H(\boldsymbol{e}_t, p; \boldsymbol{\Phi})),$$

where $\odot$ is element-wise multiplication.

In the training procedure, we have added two regularization terms. The first one is output regularizer proposed by Li & Hoiem (2017):

$$\mathcal{L}_{output} = \sum_{t=1}^{T-1} \sum_{i=1}^{|X_t|} \|f(\boldsymbol{x}_{i,t}; \boldsymbol{\theta}^* \odot \boldsymbol{m}_t) - f(\boldsymbol{x}_{i,t}; \boldsymbol{\theta} \odot \boldsymbol{m}_t)\|^2,$$

where $\boldsymbol{\theta}^*$ is the set of the target network parameters before attempting to learn task $T$.

This solution is not only expensive in terms of memory but also does not follow the online learning paradigm adequately. But hypernetworks von Oswald et al. (2019); Henning et al. (2021) avoid this problem. Task-conditioned hypernetworks produce an output depending on the task embedding. We can compare the fixed hypernetwork output produced before learning task $T$ with weights $\boldsymbol{\Phi}^*$ with the output after a current proposition of hypernetwork weight modifications $\Delta\boldsymbol{\Phi}$, according to the cross-entropy loss. The difference between HyperMask and von Oswald et al. (2019) relies on the fact that we just regularize masks dedicated to consecutive continual learning tasks and the target weights have to work in general, while von Oswald et al. (2019) regularize weights that are further directly placed in the target network.

Finally, in HyperMask, the output regularization loss is given by:

$$\mathcal{L}_{output}(\boldsymbol{\Phi}^*, \boldsymbol{\Phi}, \Delta\boldsymbol{\Phi}, \{\boldsymbol{e}_t\}) = \frac{1}{T-1} \sum_{t=1}^{T-1} \|\mathcal{H}(\boldsymbol{e}_t, 0; \boldsymbol{\Phi}^*) - \mathcal{H}(\boldsymbol{e}_t, 0; \boldsymbol{\Phi} + \Delta\boldsymbol{\Phi})\|^2,$$

where $\Delta\boldsymbol{\Phi}$ is considered fixed. We do not sparse the hypernetwork weights at this stage, i.e. $p = 0$.

Table 1: Average accuracy with a standard deviation of different continual learning methods. We obtained better results than two of our main baselines: WSN and HNET. Moreover, we have the the best results on CIFAR-100 and Tiny ImageNet and second scores in Permuted MNIST and Split MNIST. Results for different methods than HyperMask are derived from other papers.
$*$ − model trained on ResNet-20 architecture;
$**$ − model trained on ZenkeNet architecture.

| Method | Permuted MNIST | Split MNIST | Split CIFAR-100 | Tiny ImageNet |
|---|---|---|---|---|
| HAT | $\mathbf{97.67 \pm 0.02}$ | − | $72.06 \pm 0.50$ | − |
| GPM | $94.96 \pm 0.07$ | − | $73.18 \pm 0.52$ | $67.39 \pm 0.47$ |
| PackNet | $96.37 \pm 0.04$ | − | $72.39 \pm 0.37$ | $55.46 \pm 1.22$ |
| SupSup | $96.31 \pm 0.09$ | − | $75.47 \pm 0.30$ | $59.60 \pm 1.05$ |
| La-MaML | − | − | $71.37 \pm 0.67$ | $66.99 \pm 1.65$ |
| FS-DGPM | − | − | $74.33 \pm 0.31$ | $70.41 \pm 1.30$ |
| WSN, $c = 3\%$ | $94.84 \pm 0.11$ | − | $70.65 \pm 0.36$ | $68.72 \pm 1.63$ |
| WSN, $c = 5\%$ | $95.65 \pm 0.03$ | − | $72.44 \pm 0.27$ | $71.22 \pm 0.94$ |
| WSN, $c = 10\%$ | $96.14 \pm 0.03$ | − | $74.55 \pm 0.47$ | $71.96 \pm 1.41$ |
| WSN, $c = 30\%$ | $96.41 \pm 0.07$ | − | $75.98 \pm 0.68$ | $70.92 \pm 1.37$ |
| WSN, $c = 50\%$ | $96.24 \pm 0.11$ | − | $76.38 \pm 0.34$ | $69.06 \pm 0.82$ |
| WSN, $c = 70\%$ | $96.29 \pm 0.00$ | − | − | − |
| EWC | $95.96 \pm 0.06$ | $99.12 \pm 0.11$ | $72.77 \pm 0.45$ | − |
| SI | $94.75 \pm 0.14$ | $99.09 \pm 0.15$ | − | − |
| DGR | $97.51 \pm 0.01$ | $99.61 \pm 0.02$ | − | − |
| HNET+ENT | $97.57 \pm 0.02$ | $\mathbf{99.79 \pm 0.01}$ | − | − |
| HyperMask (our) | $97.66 \pm 0.04$ | $99.64 \pm 0.07$ | $\mathbf{77.34 \pm 1.94}^*$ $73.58 \pm 0.30^{**}$ | $\mathbf{76.22 \pm 1.06}^*$ |

Moreover, we have added classical $L^1$ regularization on the target network weights

$$\mathcal{L}_{target}(\boldsymbol{\theta}_t^*, \boldsymbol{\theta}_t) = \|\boldsymbol{\theta}_t^* - \boldsymbol{\theta}_t\|_1,$$

where $\boldsymbol{\theta}_t^*$ is the set of target network parameters before attempting to learn task $T$. Optionally, we can multiply $\mathcal{L}_{target}$ by the hypernetwork-generated mask (masked $L^1$) to ensure that the most important target network weights will not be drastically modified while the other ones will be more susceptible to modifications. In such a case

$$\mathcal{L}_{target}(\boldsymbol{\theta}_t^*, \boldsymbol{\theta}_t, \boldsymbol{m}_t) = \boldsymbol{m}_t \odot \|\boldsymbol{\theta}_t^* - \boldsymbol{\theta}_t\|_1.$$

During hyperparameter optimization, we compared two variants of $\mathcal{L}_{target}$, i.e. masked and non-masked $L^1$. A conclusive choice is dependent on the considered dataset.

The final cost function consists of the classical cross-entropy $\mathcal{L}_{current}$, output regularization $\mathcal{L}_{output}$, and target layer regularization $\mathcal{L}_{target}$

$$\mathcal{L} = \mathcal{L}_{current} + \beta \cdot \mathcal{L}_{output} + \lambda \cdot \mathcal{L}_{target},$$

where $\beta$ and $\lambda$ are hyperparameters that control the strength of regularization.

## 4 EXPERIMENTS

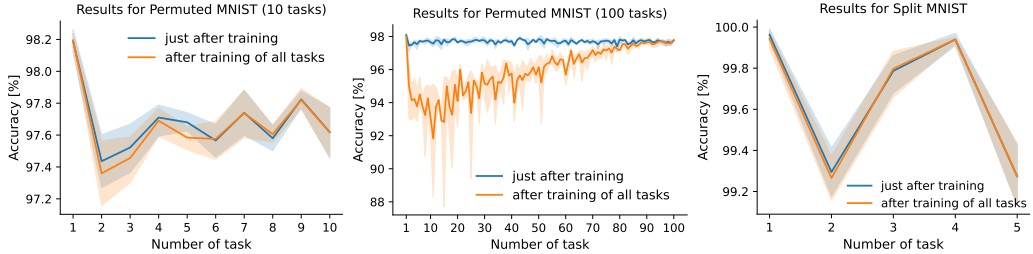

Figure 2: Visualization of mean accuracy (with 95% confidence intervals) for Permuted MNIST for 10 and 100 tasks and Split MNIST for 5 tasks. The blue lines represent test accuracy calculated after training consecutive models, while the orange lines correspond to test accuracy after finishing all CL tasks. The decrease in accuracy for 10-task Permuted MNIST and Split MNIST is very small. In the Permuted MNIST 100-task case, the mean accuracy equals $95.92 \pm 0.18$.

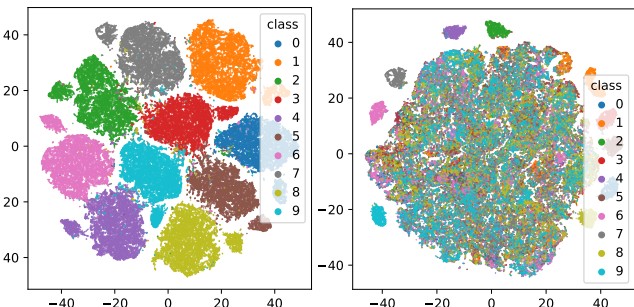

Figure 3: Visualization of a target network's output classification layer activations in two scenarios. On the left hand, we used a target network weighted by a semi-binary mask (HyperMask). On the right side, we used only the target network without a mask produced by the hypernetwork. In the first case, data sample classes are separated; in the second case, only samples from the first task are distinguished.

This section presents a numerical comparison of our model with a few baseline solutions. We analyzed task-incremental continual learning with a multi-head setup for all the experiments. We followed the experimental setups from recent works Saha et al. (2020); Yoon et al. (2020); Deng et al. (2021).

**Architecture** We used two-layered MLP with 100 neurons per layer for Permuted MNIST and Split MNIST. For Split CIFAR-100, we used ResNet-20 and ZenkeNet (Zenke et al., 2017) and for Tiny ImageNet we applied ResNet-20.

**Baselines** We compared our solution with two natural baselines: WSN Kang et al. (2022) and HNET von Oswald et al. (2019). WSN used the lottery ticket hypothesis, while HNET used the hypernetwork paradigm. We also added a comparison with strong CL baselines from different categories. In particular, we used regularisation-based methods: HAT Serra et al. (2018) and EWC Kirkpatrick et al. (2017), rehearsal-based methods like GPM Saha et al. (2020) and FS-DGPM Deng et al. (2021), a pruning-based method like PackNet Mallya & Lazebnik (2018) and SupSup Wortsman et al. (2020), and a meta learning approach like La-MAML Gupta et al. (2020).

**Experimental setting** We used the experimental setting from WSN Kang et al. (2022) and HNET von Oswald et al. (2019). We did not change the original architectures provided by the authors. Some results in the tables were directly taken from papers.

**Numerical comparison** We evaluated our algorithm on four standard benchmark datasets: Permuted MNIST, Split MNIST, Split CIFAR-100, and TinyImageNet Le & Yang (2015). In Tab. 1, we compared HyperMask with the state-of-the-art models. The most important conclusion is that we obtained better results than two of our main baselines: WSN and HNET. Moreover, we had the second score in Permuted MNIST and Split MNIST. In the case of Permuted MNIST, our exact result was equal to 97.664, so it was only 0.006 smaller than HAT. In the case of CIFAR-100, we had the best score when we used ResNet-20 and about $4\%$ less for ZenkeNet. Using ResNet-20, we outperformed all reference methods in Tiny ImageNet by over $4\%$. However, in WSN, La-MaML and FS-DPGM, authors used an architecture with four convolutional and three fully-connected layers.

**Influence of semi-binary mask on classification task** In this subsection, we show that the semi-binary mask of HyperMask helped the target network to discriminate classes in consecutive CL

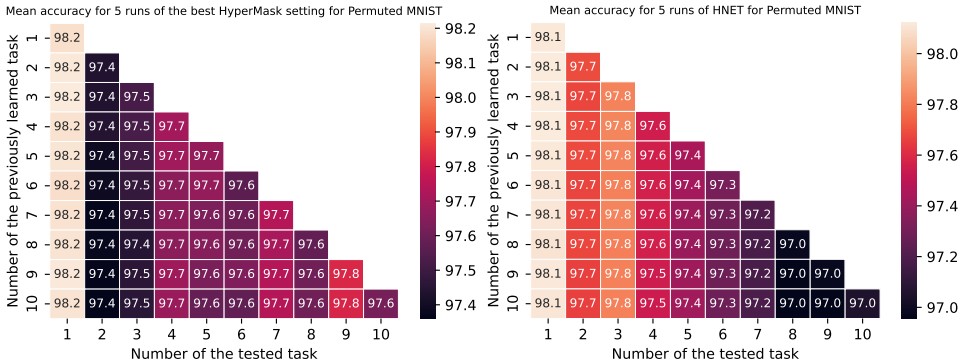

Figure 5: Mean test accuracy for consecutive CL tasks averaged over five runs of the best architecture settings of HyperMask (left side) and the default setting of HNET (right side) for ten tasks of the Permuted MNIST dataset. Training of subsequent tasks leads to a slight decrease in the overall accuracy of the previous tasks, but, in general, HyperMask achieves higher accuracy for more recent tasks while HNET is more powerful for the first tasks.

tasks. To visualize such properties, we considered the Permuted MNIST dataset (results for other datasets we included in Appendix). We took the fully-trained model and collected activations of the classification layer of the target network. In Fig. 3, we present t-SNE two-dimensional embeddings obtained from the set of activations containing all data samples from 10 tasks. Values were calculated for an exemplary model that achieved $97.72\%$ overall accuracy after 10 CL tasks. The results for a tandem hypernetwork and target network (like in HyperMask) are presented on the left side. On the right side is shown a situation in which a mask from the hypernetwork was not applied to the target network trained in HyperMask. In the first case, data sample classes are clearly separated; in the second case, only samples from the first task are distinguished. The remaining data samples form one cluster in the embedding space. Interestingly, data from the first task are separated from samples from all subsequent tasks, which indicates that the first task plays a special role for HyperMask.

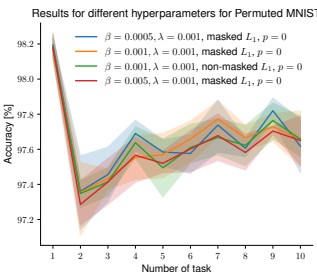

Figure 4: Visualization of stability of HyperMask. We obtained similar results for different hyperparameters.

**Forgetting of previous tasks**   The HNET models produce completely different weights for each task. In consequence, they demonstrate minimal forgetting. HyperMask models inherit such ability thanks to generating different masks for each task. To visualize such properties. we present in Fig. 2 mean accuracy (with 95% confidence intervals) for the best setting of HyperMask for ten tasks of the Permuted MNIST dataset (left side) and five tasks of the Split MNIST dataset (right side). The blue lines represent test accuracy calculated after training consecutive models, while the orange lines correspond to test accuracy after finishing all CL tasks. The decrease in accuracy is very small, and the confidence intervals almost overlap, suggesting a very limited negative backward transfer. In Fig. 5, we present a comparison of our HyperMask and HNET in terms of test accuracies for CL tasks after consecutive training sessions. Both methods suffer from performance drops only slightly. However, HyperMask is the most efficient for the first and the more recent tasks while the accuracy of HNET decreases smoothly with subsequent tasks.

Interestingly, HyperMask preserves the accuracy on the first task even after training of many subsequent ones. It is clearly visible in Fig. 2 where results for 100-task Permuted MNIST are presented. Even after training of the next 99 tasks, HyperMask has similar test accuracy on the first task to the accuracy calculated just after its training. Then, a performance drop typical for continual learning methods may be observed. It may indicate that the tandem of hyper- and the target network is getting used to the first task which strongly affects the behavior of weights.

**Stability of HyperMask model**   HyperMask models have a similar number of hyperparameters as HNET. The most critical parameters are $\beta$ and $\lambda$, which control regularization strength. We

also use a parameter describing the level of zeros in a semi-binary mask and we define whether masked or non-masked $L^1$ has to be used. Masked $L_1$ means that $\mathcal{L}_{target}$ was multiplied by the hypernetwork-generated mask while non-masked $L_1$ denotes the opposite case. In Fig. 4, we present mean test accuracy (with 95% confidence intervals) for five runs of the selected architecture settings of HyperMask, for ten tasks of the Permuted MNIST dataset, calculated after training of all tasks. The presented results indicate that a small change in hyperparameters does not cause a performance drop. The blue line represents the best hyperparameter setting found.

**Scenario with model's task prediction** We also evaluated HyperMask in a scenario in which task identity is not directly given to the model but must be inferred by the network itself. Following von Oswald et al. (2019), we prepared a task inference method based on the entropy values. After training for all tasks, consecutive data samples were propagated through the hyper- and target network for different task embeddings. The task with the lowest entropy value of the classification layer's output in the target network was selected for the final calculations. Then, the classifier decision for the corresponding embedding was considered.

Table 2: Mean overall accuracy (in %) in a scenario where the model must recognize task identity. For HNET+ENT and HyperMask, the inference is made based on the entropy results. The presented results from methods different than HyperMask are derived from von Oswald et al. (2019).

| Method | Permuted MNIST | Split MNIST |
|---|---|---|
| HNET+ENT | $91.75 \pm 0.21$ | $69.48 \pm 0.80$ |
| EWC | $33.88 \pm 0.49$ | $19.96 \pm 0.07$ |
| SI | $29.31 \pm 0.62$ | $19.99 \pm 0.06$ |
| DGR | $\mathbf{96.38 \pm 0.03}$ | $\mathbf{91.79 \pm 0.32}$ |
| HyperMask | $90.31 \pm 1.36$ | $85.80 \pm 3.08$ |

Table 2 presents results for two datasets: Permuted MNIST (10 tasks) and Split MNIST. HyperMask was compared with its natural baseline, i.e. HNET+ENT von Oswald et al. (2019), which has the same strategy adopted for task inference. Also, the results for the three different methods (EWC, SI and DGR) are shown. WSN in the paper Kang et al. (2022) only realizes the strategy in which the task identity is known in advance and the authors did not describe a method for task inference. Therefore, we did not evaluate WSN in the above scenario.

The results indicate that, in this strategy, for the two datasets presented, the most competitive method is DGR. Our main baseline, HNET+ENT, is slightly better than HyperMask for Permuted MNIST (with 10 tasks) and considerably worse for the Split MNIST dataset. For HyperMask, we also calculated mean task prediction accuracy, which is equal to $90.30 \pm 1.56$ for Permuted MNIST and $62.90 \pm 5.83$ for Split MNIST. The discussed scores indicate a potential of HyperMask for task inference approaches, i.e. with another neural network for task prediction.

**Limitations and future works** One of the main limitations of HyperMask is the memory consumption due to the fact that the hypernetwork output layer must have the same number of neurons as the number of parameters in the target network. The chunking approach described in von Oswald et al. (2019), in which the target's weight values are generated by the hypernetwork partially, was not adopted in HyperMask because it led to considerably worse results so far. However, this approach should be analyzed thoroughly and may bring positive future results.

HyperMask may be considered in few-shot class incremental learning in which a model is trained in a large number of base samples and then a small portion of samples representing new classes is delivered to the model Kang et al. (2023). Due to the high accuracy of HyperMask on the first task (despite many subsequent ones), our method may be very useful in this CL scenario.

## 5 CONCLUSION

We present HyperMask, a method that trains a single network for all tasks. The hypernetwork produces semi-binary masks to generate target subnetworks tailored to new tasks. This approach utilizes the hypernetwork's capacity to adjust to new tasks with minimal forgetting. Also, due to the lottery ticket hypothesis, we can use a single network with weighted subnets devoted to each task.

The experimental section shows that our model performs better than lottery ticket and hypernetwork -based continual learning models. We also obtained comparable results to the state-of-the-art methods. We applied our method for multilayer perceptions and convolutional neural networks working as classifiers. HyperMask also has a potential for application in strategies in which task identity has to be inferred by the method and is not known a priori.

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

## A APPENDIX: BACKWARD TRANSFER

We used overall accuracy for the evaluation of backward transfer (BWT) for the selected CL methods: WSN Kang et al. (2022), HNET von Oswald et al. (2019) and HyperMask. BWT measures forgetting previous tasks after learning the subsequent ones:

$$BWT = \frac{1}{T-1} \sum_{i=1}^{T-1} A_{T,i} - A_{i,i},$$

where $A_{T,i}$ is the test accuracy for task $i$ after training on task $T$, while $A_{i,i}$ is the test accuracy for task $i$ just after training the model on this task. Negative BWT means that learning new tasks caused the forgetting of past tasks. Zero BWT represents a situation where the accuracy of CL tasks did not change after learning the new knowledge. Finally, positive BWT corresponds to the state in which the model gained additional knowledge after learning the next tasks that improved the accuracy of the previous CL tasks.

Table 3 presents mean backward transfer for 5 training runs of HNET+ENT and HyperMask for three experiments: Permuted MNIST with 10 or 100 tasks and Split MNIST. By definition, WSN remembers masks from the preceding tasks. Therefore, the backward transfer in this case is always equal to zero. HNET and HyperMask achieved comparable and slightly negative values of BWTs for Permuted MNIST on 10 tasks and Split MNIST. In the case of Permuted MNIST on 100 CL tasks, we only have results for HyperMask. Despite a much larger number of tasks, the negative backward transfer did not exceed 2%, which means that HyperMask is largely immune to catastrophic forgetting.

Table 3: Mean backward transfer (in %) with standard deviation for different continual learning methods.

| Dataset | Permuted MNIST | | Split MNIST |
|---------|----------------|-----------|-------------|
| | 10 tasks | 100 tasks | |
| WSN, $c = 30\%$ | 0.0 | − | − |
| HNET+ENT | $-0.018 \pm 0.01$ | − | $-0.027 \pm 0.07$ |
| HyperMask | $-0.025 \pm 0.03$ | $-1.791 \pm 0.18$ | $-0.009 \pm 0.04$ |

## B  APPENDIX: ARCHITECTURE DETAILS

We implemented HyperMask in Python 3.7.16 with the use of such libraries like hypnettorch 0.0.4 von Oswald et al. (2019), PyTorch 1.5.0, NumPy 1.21.6, Pandas 1.3.5, Matplotlib 3.5.3, seaborn 0.12.2 and others. All network training sessions were performed using several NVIDIA GeForce RTX 2080 Ti graphic cards.

We tried to implement hypernetwork / target network architectures close to the work presenting HNET algorithm von Oswald et al. (2019), but for some hyperparameters, especially those present only in HyperMask, we performed an intensive grid search optimization. In all cases, we did not use chunked hypernetworks, i.e. we did not generate a mask in small pieces. It means that the hypernetwork output layer always had such a number of neurons as the number of weights of the target network. This is due to the fact that the size of the generated mask has to be the same as the number of target network parameters. This solution is more memory expensive than the chunking approach but it ensures higher classification accuracy in the case of HyperMask.

**Permuted MNIST**  Final experiments on the Permuted MNIST dataset with 10 CL tasks were performed using the following architecture. The hypernetwork had two hidden layers with 100 neurons per each. As the target network was selected a multilayer perceptron with two hidden layers of 1000 neurons and ELU activation function with $\alpha$ hyperparameter regarding the strength of the negative output equaling to 1. The size of the embedding vectors was set to 24. The sparsity parameter $p$ was adjusted to 0 and the regularization hyperparameters were as follows: $\beta = 0.0005$ and $\lambda = 0.001$. Furthermore, a masked $L^1$ regularization was chosen. The training of models was performed through 5000 iterations with a batch size of 128 and Adam optimizer with a learning rate set to 0.001. Finally, models after the last training iterations were selected. The validation set consisted of 5000 samples. The data was not augmented. The presented results are averaged over 5 training runs for different seed values. Also, the dataset was padded with zeros and the final size of the MNIST images was $32 \times 32$.

For 100 CL tasks, the hyperparameters were the same as above, but 3 training runs were performed.

To select the best hyperparameter set, we performed an intensive hyperparameter optimization. In the final stage, we evaluated, in different configurations, various hypernetwork set-

tings ([25, 25], [100, 100]), masked and non-masked $L^1$ regularization, $p \in \{0, 30\}$, $\beta \in \{0.0005, 0.001, 0.0025, 0.005, 0.01, 0.1\}$ and $\lambda \in \{0.0005, 0.001, 0.0025, 0.005, 0.01\}$.

In the initial experiments, we also considered hypernetworks having 2 hidden layers with 50 neurons per each, embeddings of sizes 8 and 72, a learning rate of $0.0001$, batch size of 64, $\beta = 0.05$, $\lambda \in \{0.0001, 0.00001, 0.05\}$ and $p = 70$.

**Split MNIST**  For this dataset with 5 CL tasks, we applied data augmentation and trained models through 2000 iterations. The best-performing model was composed of a hypernetwork with two hidden layers with 25 neurons per each and a target network with two hidden layers consisting 400 neurons. We used $\beta = 0.001$ and a sparsity parameter $p = 30$. Furthermore, the embedding size was 128. In each task, 1000 samples were assigned to the validation set. The rest of the hyperparameters were exactly the same as for the Permuted MNIST, i.e. we applied a masked $L^1$ regularization with $\lambda = 0.001$, ELU activation function with $\alpha = 1$, Adam optimizer with a learning rate of $0.001$ and batch size of 128. Also, the mean results are averaged over 5 training runs.

During the hyperparameter optimization stage, we evaluated models with embedding sizes of $24, 72, 96$ and $128$, hypernetworks with hidden layers of shapes $[10, 10], [25, 25]$ and $[50, 50]$, masked and non-masked $L^1$ regularization, batch sizes of 64 and 128, $\beta \in \{0.001, 0.01\}$, $p \in \{0, 30, 70\}$ and $\lambda \in \{0.0001, 0.001\}$.

**CIFAR-100**  In this dataset, we assumed 10 tasks with 10 classes per each. Another version of this CL benchmark adopts CIFAR-10 and 5 tasks (i.e., 50 classes) of CIFAR-100 dataset, like in von Oswald et al. (2019). However, we selected the first scenario, similarly as in Kang et al. (2022).

We performed experiments for two different convolutional target networks: ResNet-20 and ZenkeNet. The first of them was similar to the network considered in von Oswald et al. (2019) but it was slightly shorter, while the second one was more similar to AlexNet used in Kang et al. (2022). ZenkeNet was even a less sophisticated architecture than AlexNet.

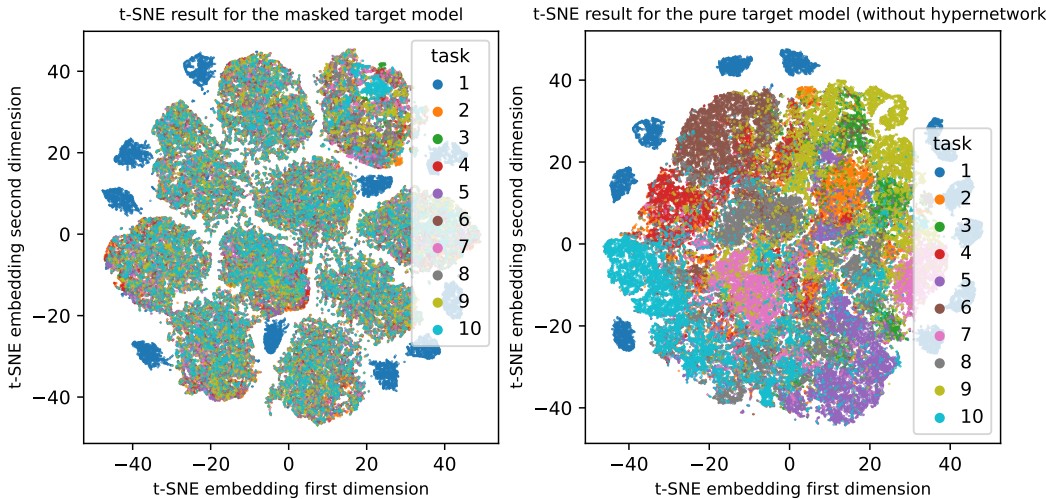

Figure 6: t-SNE embeddings of features extracted from all data samples of 10 tasks of the Permuted MNIST dataset, created similarly as presented in Fig. 3, but the samples are labelled relative to the CL tasks. On the left column, results for HyperMask (i.e. hypernetwork and target network) are shown. On the right column are presented only results for the target network, without the application of a mask from hypernetwork. The plots clearly indicate that samples from the first task form a separate structure in the data space. Even when the classical version of HyperMask is used, the first task plays a particular role.

In more detail, we selected a ResNet architecture containing 20 layers with 9 residual blocks and a widening factor equal to 2, which means doubling the convolutional filters. During the hyperparameter optimization, we also considered a narrower architecture as well as shorter and longer ResNets (up to 32 layers) but they were less promising than a 20-layer network. Also, batch nor-

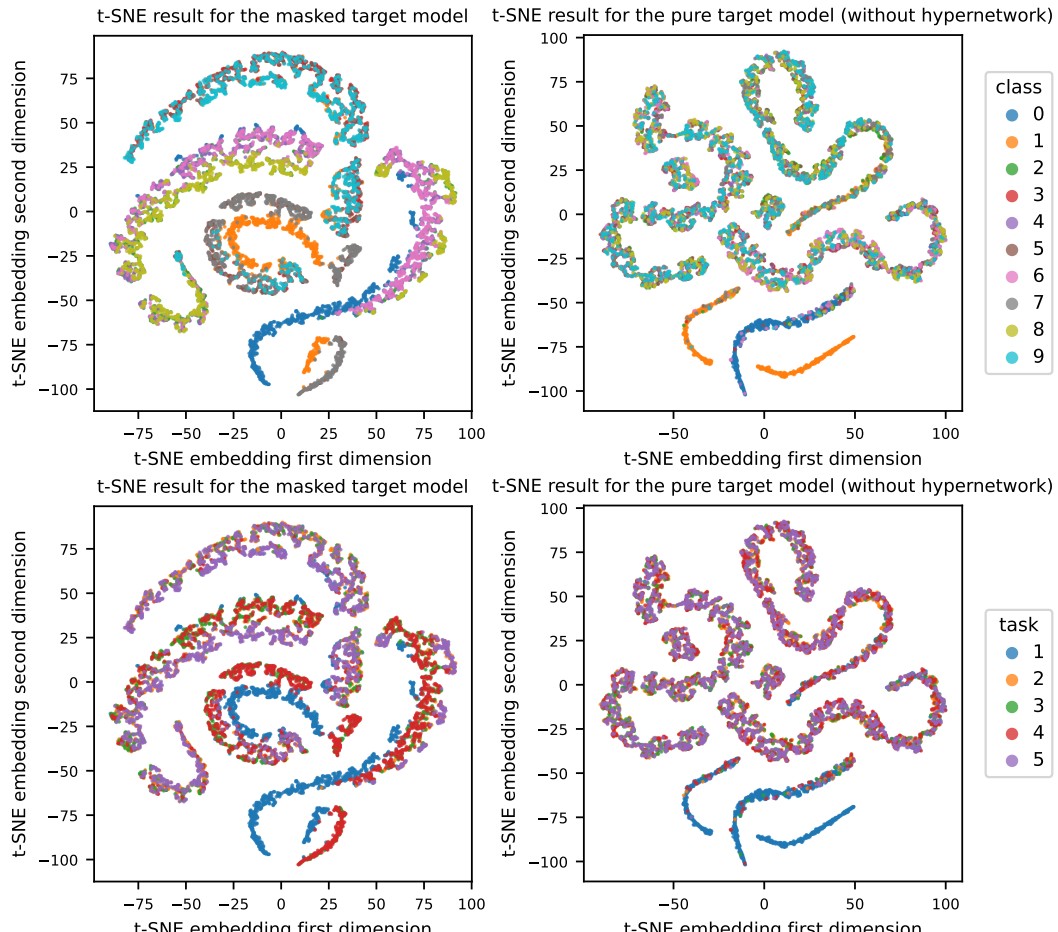

Figure 7: t-SNE embeddings of features extracted from all data samples of 5 tasks of the Split MNIST dataset. Values were taken from the classification layer of the target network for an exemplary model that achieved $99.76\%$ overall accuracy after 5 CL tasks. On the left column, results for a tandem hypernetwork and target network (like in HyperMask) are presented. On the right column is shown a mask from hypernetwork that was not applied to the target network while it trained in tandem in HyperMask. In the first case, classes form different clusters, especially the pairs of classes that were mutually compared in consecutive CL tasks (0 and 1, 2 and 3, etc.). In the second case, only 0's and 1's are separated while the remaining data samples are mixed in the embedding subspace. Furthermore, in this situation, data from the first task form a separate cluster which suggests that it mainly defines the structure of the data space.

malization was used (these layers were excluded from multiplying by hypernetwork-based masks). Batch statistics were calculated even during the evaluation, i.e. parameters were not stored after consecutive CL tasks.

ZenkeNet was a convolutional neural network described in Zenke et al. (2017). It consisted of two blocks of two convolutional layers containing 32 and 64 filters, respectively. Each block was finished by a single max pooling layer. Finally, the network had two fully connected layers with 512 and 10 neurons, respectively.

During the hyperparameter optimization for ZenkeNet, we compared models having embedding size set to 48, a hypernetwork with one hidden layer with 100 neurons, trained with Adam optimizer using a learning rate of 0.001 and batch size of 32. A non-masked $L^1$ regularization was selected as the more promising. Furthermore, we evaluated $\beta \in \{0.01, 0.1, 1\}$ and $\lambda \in \{0.01, 0.1, 1\}$. The dataset was augmented, and in the validation set were 500 samples. Furthermore, the sparsity parameter $p$ was set to 0. It is worth emphasizing that the lack of data augmentation led to significantly lower

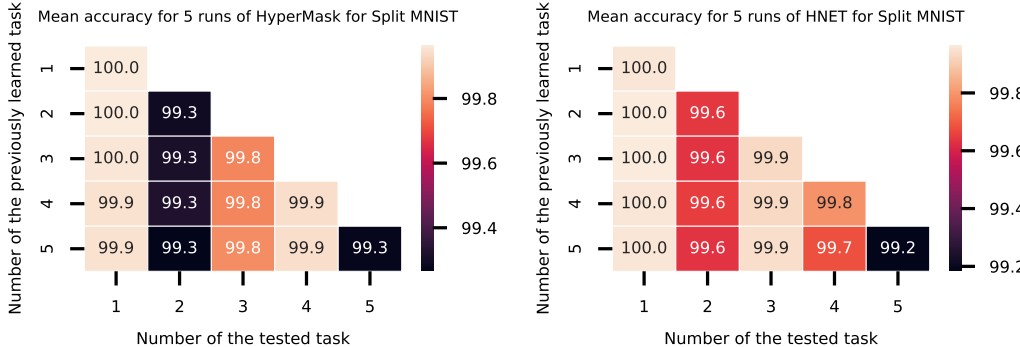

Figure 8: Mean test accuracy for consecutive CL tasks, averaged over 5 runs of the best architecture settings of HyperMask (left side) and the default setting of HNET (right side) for 5 tasks of the Split MNIST dataset. Similarly, as for the Permuted MNIST, the performance drops after consecutive CL tasks are slight. Interestingly, the second and the last tasks (i.e., classification of numbers 2 and 3 or numbers 8 and 9, respectively) are the most challenging for both methods.

accuracy than in the opposite case. The network training was performed through 200 epochs and the best model was chosen based on the validation loss. Also, a learning rate scheduler was applied, i.e. after five consecutive epochs without improvement, the learning rate was multiplied by $\sqrt{0.1}$. In the case of ZenkeNet, for the final experiments, $\beta = 0.01$ and $\lambda = 0.01$ were selected.

For the ResNet, most of the hyperparameters were the same as for ZenkeNet, excluding $\beta = 0.01$ and $\lambda = 1$.

**Tiny ImageNet** In this case, we divided the dataset randomly into 40 tasks with 5 classes, similarly to Kang et al. (2022). We assumed the same training strategy, i.e. we learned each task through 10 epochs and the validation set consisted of 250 samples. We performed experiments with ResNet-20 architecture, which is explained in detail in the section devoted to CIFAR-100. However, in WSN Kang et al. (2022), La-MaML Gupta et al. (2020) and FS-DPGM Deng et al. (2021) authors used an architecture with four convolutional and three fully-connected layers. The best models were chosen according to the values of the validation loss. We augmented the dataset using random cropping and horizontal flipping. For the hypernetwork, we selected a multilayer perceptron with two hidden layers of 100 neurons while the size of the task embedding vector was set to 96. Furthermore, we used non-masked $L^1$ regularization with $\beta = 1$ and $\lambda = 0.1$. Also, the sparsity parameter $p$ was adjusted to 0 in this case. We performed training with Adam optimizer with batch size set to 16 and learning rate set to 0.0001. The learning rate scheduler was exactly the same as for CIFAR-100, i.e. a patience step was equal to 5 epochs and the multiplication factor was $\sqrt{0.1}$.

In the hyperparameter optimization stage, we considered an order of magnitude greater learning rate, i.e. 0.001, like in Kang et al. (2022), but it led to weaker results. Also, other sizes of embedding vectors, consisting of 48 and 128 coordinates, were further from optimal solutions than 96. Similarly, smaller hypernetworks, i.e., those consisting of two hidden layers with 10 neurons or one hidden layer with 100 neurons, were rejected. We also tested a masked $L^1$ regularization and lower values of $\beta$ and $\lambda$ but stronger regularization is preferred for this target network architecture and such a complicated dataset. For some solutions, $p = 30$ led to better solutions than $p = 0$, but finally, we selected $p = 0$.

## C APPENDIX: INFLUENCE OF SEMI-BINARY MASK ON CLASSIFICATION TASK

In the main paper, we showed that the semi-binary mask of HyperMask helps the target network to discriminate classes in consecutive CL tasks. We considered the Permuted MNIST dataset to visualize such properties (see Fig. 3). Now, in Fig. 6, we present plots for this dataset with data samples labelled according to the CL task and present results on Split MNIST; see Fig. 7. Interestingly, even when the mask was not applied, the first task was still solved correctly and the corresponding samples formed separate clusters. Furthermore, data from the first task form a separate structure in the

embedding subspace, even when only the target network is applied. This suggests that the first task is especially important. This observation is supported by the results presented in Fig.2, where the classification accuracy for the first task remains high despite learning many subsequent CL tasks.

## D  APPENDIX: FORGETTING OF PREVIOUS TASKS

The HNET model produces completely different weights for each task. In consequence, it demonstrates minimal forgetting. HyperMask model inherits such ability to minimize forgetting previous tasks thanks to masks created by hypernetworks. To visualize such properties, we used the Permuted MNIST dataset, see Fig. 5, to compare HyperMask with HNET. Now we show analogical results on Split MNIST in Fig. 8. Both methods feature minimal forgetting after training of consecutive CL tasks. The situation changes when we consider a more demanding dataset like Split CIFAR-100, see Fig. 9. ZenkeNet, despite lower classification accuracy than ResNet-20, achieved only a slight decrease in performance after subsequent tasks. In the case of ResNet-20, the drop in efficiency was considerable, for instance from $83.1\%$ just after learning of the first task to $73.7\%$ at the end of the CL scenario. However, a more intense regularization (i.e. higher values of $\beta$ and $\lambda$) which may prevent the network from knowledge forgetting, led to slightly lower accuracy averaged over 10 tasks. Therefore, a final hyperparameter choice has to be a compromise.

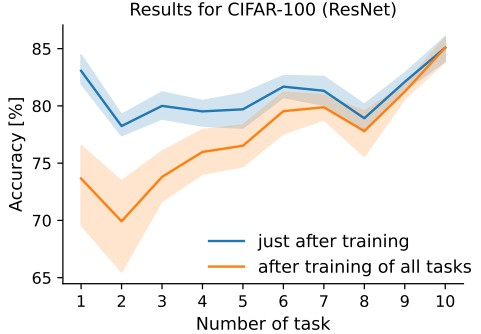
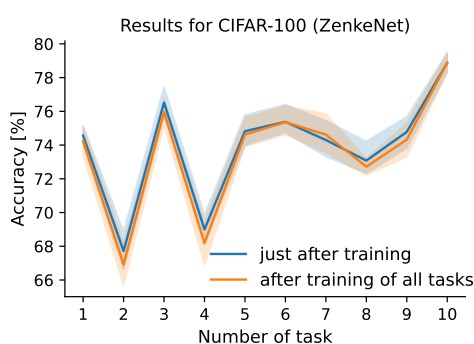

Figure 9: Visualization of mean accuracy (with $95\%$ confidence intervals) of HyperMask for 10 tasks of the CIFAR-100 dataset for two different target network architectures (ResNet-20 and ZenkeNet). The blue lines represent test accuracy calculated after training subsequent models, while the orange lines correspond to test accuracy after finishing training for all CL tasks. Despite the fact that ResNet-20 achieved higher accuracy than ZenkeNet, it suffers from catastrophic forgetting in a more severe way than the second considered architecture.

## E  APPENDIX: STABILITY OF HYPERMASK MODEL

HyperMask model has a similar number of hyperparameters as HNET. The most critical parameters are $\beta$ and $\lambda$, which control regularization strength. This method also has parameter $p$ describing the level of zeros in consecutive layers of the semi-binary mask. Moreover, we define whether $L^1$ will be multiplied by the mask values or not. Furthermore, similarly as in HNET, there exists another branch of hyperparameters regarding the networks' shape, for instance, the hypernetwork embedding size, the number of hidden layers and the number of neurons in consecutive layers. Similarly, we have to define the setting of the target network.

In Fig. 11, we present mean test accuracy for consecutive CL tasks averaged over 2 runs of different architecture settings of HyperMask for 5 tasks of the Split MNIST dataset. Most of the HyperMask models achieve the highest classification accuracy for the first CL task while the weakest one for the subsequent task. In all of the above plots, results are compared with the best hyperparameter setting, i.e. embedding size is equal to 128, hypernetwork has two hidden layers with 25 neurons per each, $\beta = 0.001$, $\lambda = 0.001$, $p = 30$, the batch size is equal to 128 and the masked $L^1$ norm is applied. In consecutive subplots, some of the above hyperparameters are changed and the performance of corresponding models is compared with the most efficient setup.

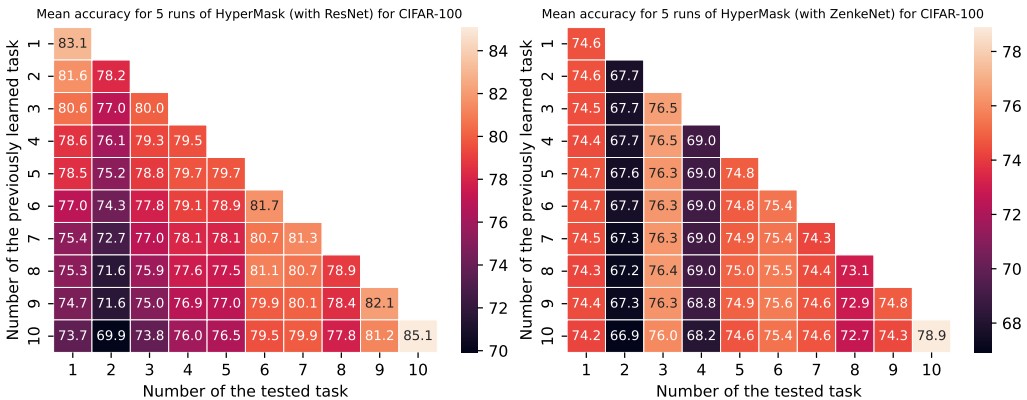

Figure 10: Mean test accuracy for consecutive CL tasks averaged over five runs of HyperMask for two target network architectures: ResNet-20 (left side) and ZenkeNet (right side) for ten tasks of the CIFAR-100 dataset. For ZenkeNet only a slight decrease in overall accuracy for previous tasks may be noticed while for ResNet-20 there is a substantial drop in performance. However, the mean results are better for ResNet-20, regardless.

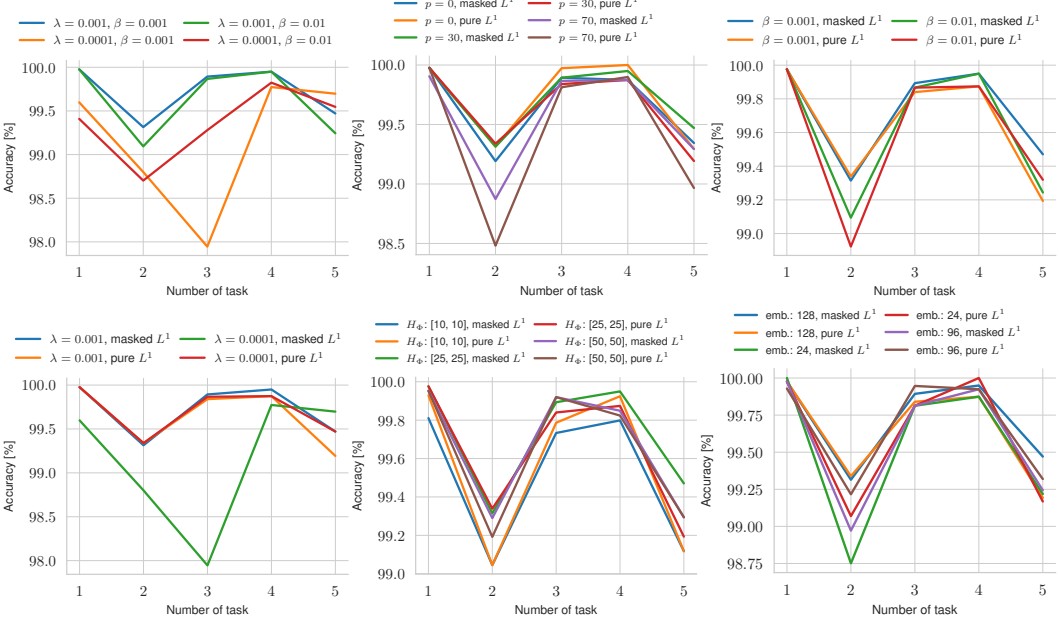

Figure 11: Mean test accuracy for consecutive CL tasks averaged over 2 runs of different architecture settings of HyperMask for 5 tasks of the Split MNIST dataset. Most of the HyperMask models achieve the highest classification accuracy for the first CL task while the weakest one for the subsequent task. In many cases, differences in performance of the compared models are small.

## F   APPENDIX: TIME CONSUMPTION

We depicted in Tab. 4 the mean training time of HyperMask for five tasks of Split MNIST, ten tasks of Permuted MNIST and ten tasks of Split CIFAR-100 using a single NVIDIA GeForce RTX 2080 Ti graphic card. For the easiest dataset, HyperMask needs only slightly more than 21 minutes. In the case of Permuted MNIST, which consists of more advanced CL tasks, HyperMask needs less than 2 hours. For Split CIFAR-100 and more complicated convolutional architectures, calculation times are higher: about 6 hours for ZenkeNet and more than 10 hours for ResNet-20.

Table 4: Mean training time of HyperMask for different datasets.

| Dataset | Mean calculation time in HH:MM:SS (with standard deviation) |
| --- | --- |
| Split MNIST | $00:21:06 \pm 00:02:37$ |
| Permuted MNIST | $01:45:14 \pm 00:04:07$ |
| Split CIFAR-100 (ZenkeNet) | $06:02:25 \pm 00:01:54$ |
| Split CIFAR-100 (ResNet-20) | $10:26:37 \pm 00:10:05$ |

