# OpenReview forum: "HyperMask: Adaptive Hypernetwork-based Masks for Continual Learning"
_ICLR.cc/2024/Conference — Submitted to ICLR 2024_

### Official Review · Reviewer_1qn2 · 2023-10-31

**Soundness:** 2 fair
**Presentation:** 2 fair
**Contribution:** 1 poor
**Rating:** 3
**Confidence:** 5

**Summary:**

The authors are proposing a CIL method that can make use of a single network for multiple tasks. Based on the lottery ticket hypothesis, the proposed method only modifies the weights related to the specific task and this preserves the performance of previous tasks. Even though the hyper network consumes extra memory, it can keep the base network from growing over tasks.

**Strengths:**

- The method itself is technically reasonable and the details of the method are well described in the paper.

**Weaknesses:**

- Baselines suggested in this paper are relatively outdated. I understand that incremental architecture methods are no longer dominating this field of research but it does not mean that they can ignore regularization or representation based methods. For example, FeCAM [1] does not even need to know the task index and it still outperforms the author's method.
- The amount of experimental result is severely insufficient. According to the paper, HyperMask is superior to other methods only in Split-Cifar100 and in my knowledge this is not the best one in continual learning. Please refer to [1].
- Almost all techniques utilized in this paper are brought from previous works and I'm not sure if this is something new in CIL. Also if the authors were to mention about Lottery-ticket hypothesis, I think there should be something more than this. This method looks like a combination of HAT and PackNet to me.

[1] Goswami, Dipam, et al. "FeCAM: Exploiting the Heterogeneity of Class Distributions in Exemplar-Free Continual Learning." arXiv preprint arXiv:2309.14062 (2023).

**Questions:**

- I would be appreciated if the comments above are resolved.

---

> ### Author Response · Authors · 2023-11-17
> **Response to Reviewer 1qn2, part 1/3**
>
> We thank the reviewer for the comments related to our manuscript. We refer to your remarks in the text below.
>
> *Q1*: Baselines suggested in this paper are relatively outdated. I understand that incremental architecture methods are no longer dominating this field of research but it does not mean that they can ignore regularization or representation based methods. For example, FeCAM [1] does not even need to know the task index and it still outperforms the author's method.
>
> *R1*:
> * **HyperMask lies in a completely different category of continual learning methods**. Therefore, its comparison with FeCAM is not trivial. FeCAM is dedicated to a different setting where we have a pre-trained backbone, which does not change during the next training iterations. Authors use pre-trained architectures, like ResNet-18 and ViT-B/16, to generate class prototypes. They model the feature covariance relations using Mahalanobis distance to determine decision boundaries. FeCAM is not an architecture-based method since the architecture is not changed during training. We want to highlight that a setting with a frozen backbone and prototypes is important but is not dedicated to architecture-based approaches. Nevertheless, FeCAM is a good benchmark for the prototypical methods.
>
> * Consequently, **no experiment from our paper matches any experiment in FeCAM**. By comparing the results from HyperMask and FeCAM, it is impossible to say which model works better because the evaluation strategies of the methods are different. Authors of FeCAM state, as follows:
> > We experiment with three different incremental settings for CIFAR100 and ImageNet-Subset: 1) 50 initial classes and 5 incremental learning (IL) tasks of 10 classes; 2) 50 initial classes and 10 IL tasks of 5 classes; 3) 40 initial classes and 20 IL tasks of 3 classes. For TinyImageNet, we use 100 initial classes and distribute the remaining classes into three incremental settings: 1) 5 IL tasks of 20 classes; 2) 10 IL tasks of 10 classes; 3) 20 IL tasks of 5 classes.
>
> * In HyperMask, **we followed the experimental scenarios from architecture-based methods like HNET [1] and WSN [2]**. In the case of CIFAR-100, the dataset was divided into 10 tasks with 10 classes per task, as in [2]. Similarly, we added experiments for TinyImageNet according to the strategy described in [2] or [3], i.e. we created 40 tasks with 5 classes in a single task. It means that the evaluation criteria are different and there is no possibility of comparing the results for the experiments mentioned above.
>
> * We also think that models that do not use a pre-train backbone are still very important. We compare our model with **WSN (published in ICML 2022)** in a similar setting to the depicted one. We also compare our results with two strong algorithms: La-MaML [3] from 2020 and FS-DGPM [4] from 2021.
>
> * In the paper, in Table 2, **we described results for a continual learning setting in which the task identity has to be recognized by the model itself**. The results are presented for Permuted MNIST and Split MNIST datasets. The method applied was very simple because it was based on the entropy calculation, and still, the results were competitive. It suggests that the application of the more advanced approach (for instance, based on clustering methods used for the target network classification layer) should lead to high performance. It is part of our future work.
>
> [1] J. von Oswald et al., Continual learning with hypernetworks, ICLR 2020.
>
> [2] H. Kang et al., Forget-free Continual Learning with Winning Subnetworks, ICML 2022.
>
> [3] G. Gupta et al., La-MAML: Look-ahead Meta Learning for Continual Learning, Advances in NeurIPS, 2020.
>
> [4] D. Deng et al., Flattening sharpness for dynamic gradient projection memory benefits continual learning, In Advances in NeurIPS, 2021.

---

> ### Author Response · Authors · 2023-11-17
> **Response to Reviewer 1qn2, part 2/3**
>
> *Q2*: The amount of experimental result is severely insufficient. According to the paper, HyperMask is superior to other methods only in Split-Cifar100 and in my knowledge this is not the best one in continual learning. Please refer to
> > Goswami, Dipam, et al. "FeCAM: Exploiting the Heterogeneity of Class Distributions in Exemplar-Free Continual Learning." arXiv preprint arXiv:2309.14062 (2023).
>
> *R2*:
> * We performed experiments on TinyImageNet, a demanding dataset consisting of 40 continual learning tasks with 5 classes. A single image has a shape of 64 x 64 x 3. We prepared experiments using ResNet-20 architecture with an embedding vector of 96, β and λ equalling 1 and 0.1, respectively. Furthermore, p was set to 0, batch size to 16 and learning rate to 0.0001. Similar to WSN, the number of training epochs for each task was 10, and 50 validation samples from each class were selected in each CL task.
>
> * **We achieved state-of-the-art results, outperforming all reference methods.** The mean classification accuracy for 5 runs of HyperMask was equal to **76.22%**, while the best baseline, WSN, reached 71.96% using the architecture with 4 convolutional and 3 fully connected layers. Full results for **TinyImageNet** are depicted in the following table:
>
> ## Results for TinyImageNet
> | Method        | Mean classification accuracy in % (with std dev.) |
> |---------------|---------------------------------------------------|
> | La-MaML [3]   | 66.90 +/- 1.65                                    |
> | GPM           | 67.39 +/- 0.47                                    |
> | FS-DGPM [4]   | 70.41 +/- 1.30                                    |
> | PackNet       | 55.46 +/- 1.22                                    |
> | SupSup        | 59.60 +/- 1.05                                    |
> | WSN, c = 3%   | 68.72 +/- 1.63                                    |
> | WSN, c = 5%   | 71.22 +/- 0.94                                    |
> | WSN, c = 10%  | 71.96 +/- 1.41                                    |
> | WSN, c = 30%  | 70.92 +/- 1.37                                    |
> | WSN, c = 50%  | 69.06 +/- 0.82                                    |
> | **HyperMask** | **76.22 +/- 1.06**                                |

---

> > ### Author Response · Authors · 2023-11-17
> > **Response to Reviewer 1qn2, part 3/3**
> >
> > *Q3*: Almost all techniques utilized in this paper are brought from previous works and I'm not sure if this is something new in CIL. Also if the authors were to mention about Lottery-ticket hypothesis, I think there should be something more than this. This method looks like a combination of HAT and PackNet to me
> >
> > *R3*:
> > 1. In our opinion, HyperMask is not a combination of HAT and PackNet. **In HAT and PackNet, we do not have hypernetworks or any meta-models**.
> >
> > * In PackNet, a new subset of network weights is selected during training consecutive tasks. Also, all weights filtered for a given task must remain unchanged in the following tasks and cannot be pruned in the future. However, the number of available weights, which were not considered important for the previous tasks, is getting smaller according to the increasing number of tasks. Therefore, the space for modifying weights for some last tasks may be relatively low. In HyperMask, **we can create any number of embeddings** for the hypernetwork. Also, **the target network weights are filtered dynamically**, i.e. for the i-th task, a given weight may be considered as significant, while for the (i+1)-task, it can be irrelevant and again, for the (i+2)-task, it may be important.
> > Furthermore, except for sparsifying the target network weights, they are **scored continuously**. They do not have to be just switched on or off. Therefore, this concept is different than in the case of PackNet.
> >
> > * The mechanism of HAT is completely different than in HyperMask. **HAT uses an attention mechanism and dynamically creates or destroys paths between single network units**. Furthermore, a cumulative attention vector considering knowledge from previous tasks is created during the learning of consecutive tasks. In our approach, masks are produced by the hypernetwork depending on the embedding vector that is also trained during the learning process. Thus, **none of the above algorithms (HAT and PackNet) uses a meta-model for learning**, while in HyperMask, we have a hypernetwork producing masks for another network model adapted to the given task.
> >
> > 2. HyperMask combines two continual learning (CL) methodologies and provides a new concept using a meta-model to produce subnetworks. **The masks produced by the hypernetwork modulate the weights of the main network and act like dynamic filters, enhancing the target weights important for a given task and decreasing the importance of the remaining weights.**
> >
> > ### Architecture-based CL methods
> > + To the best of our knowledge, our model is the first architecture-based CL model that uses hypernetwork, or, in general, any meta-model, for producing masks for other networks.
> > + Hypernetwork is a meta-model that builds masks by sharing information across subsequent tasks in the meta-model’s weights. Therefore, knowledge from previous tasks is stored in the hypernetwork weights and the masks produced are not completely independent.
> > + In consequence, we do not freeze any target model weights like in many architecture-based models.
> >
> > ### Hypernetwork-based methods
> > + A single hypernetwork model produces sets of weights dedicated to each CL task. **Updates of hypernetworks are prepared not directly for the weights of the main model, like in HNET [1], but for masks modulating the target model.**
> > + HyperMask, unlike most of the methods used, produces trainable updates for a trainable model without freezing in any part.
> > + We show that hypernetworks may be used completely differently than classical hypernetwork CL models like HNET [1].

---

> ### Comment · Reviewer_1qn2 · 2023-11-21
> **Response to authors**
>
> Thanks for the prompt response of the authors. Here are several issues I still have.
>
> 1.
> - FeCAM can be used upon a pre-trained network but the main method described in the paper is training on the first task and keep it frozen during the remaining training phases. Therefore it's not a method that makes use of something more than this paper.
> - I do know the difference of experimental settings but the one from FeCAM is **far more difficult** because the task index is not given during the evaluation. Furthermore, FeCAM has demonstrated on a smaller architecture and it's hard to find the benefit of this method then.
> - But as the authors say, considering the difference on settings and setting limits on the range of baselines, the performance seems to be reasonable
>
> 2.
> - I think it's a difference of a perspective but in my opinion, HAT trains individual embedding vectors for each task through a learnable module and I don't see the difference from the hypernetwork in terms of controlling the main network from outside. Accordingly, the contibutions the authors are claiming does not seem to be valid.

---

> > ### Author Response · Authors · 2023-11-21
> > **Response to Reviewer 1qn2**
> >
> > *Ad. 1.*
> > FeCAM must store prototypes for each task similar to our emerging. Furthermore, **our model also works in settings without a known task index and results for such experiments are presented in Table 2**. But still, we claim that no experiment from our paper can be compared with any experiment from FeCAM. Such papers are not comparable since they work in different settings and continual learning tasks were prepared there differently.
> >
> > *Ad. 2.*
> > In practice, HAT is done completely differently. Authors use a task-based hard attention mechanism that maintains the information from previous tasks without affecting learning a new task. Concurrently to learning a task, they also learn almost binary attention
> > vectors through gated task embeddings, using backpropagation and minibatch stochastic gradient descent (SGD). **HyperMask works in a completely different way because it uses hypernetwork (metamodel).** We obtain better results than HAT. Therefore, we claim that our method was experimentally proven to be better for controlling network weights.
> >
> > If you feel your concerns have been addressed, please kindly consider if it is possible to update your score.

---

> > > ### Comment · Reviewer_1qn2 · 2023-11-22
> > > **Response to authors**
> > >
> > > - Thanks for correcting me about not using the task index during inference. However if so, there is even no reason to exclude FeCAM for comparison. In my perspective, HyperMask and FeCAM are just solving CIL without task index and methods such as HAT and PackNet should not be in the same category since they require task index.
> > > - I still think the experiment result is not enough to convince me and I leave the score as is.

---

> > > > ### Author Response · Authors · 2023-11-22
> > > > **Response to Reviewer 1qn2**
> > > >
> > > > Dear Reviewer,
> > > >
> > > > Thank you for your time and consideration of our work.
> > > >
> > > > *1.*
> > > >
> > > > Indeed, we do not set HAT and PackNet in the same category when we do not deliver the task index. In Table 2, where we present results in the setting in which the number of tasks must be identified by the model, HAT and PackNet are omitted. They are included in Table 1 when the results with a known task are presented.
> > > >
> > > > *2.*
> > > > - The experimental setting of FeCAM is completely different than HyperMask because, in FeCAM, a backbone consists of a ViT encoder pre-trained on ImageNet-21K, while in HyperMask, we trained our models from scratch. Therefore, a fair comparison of these approaches is not possible. Despite this fact, in Split CIFAR-100, FeCAM achieved 85.7% of classification accuracy, while in the case of HyperMask, it was 77.34%. Taking into account that we used a convolutional neural network as a backbone (not a transformer) and that we did not apply a transfer learning approach, the difference does not seem to be major. We emphasize that the CL settings and the way of training are different.
> > > >
> > > > - According to our experiments in TinyImageNet, which we additionally computed, we divided the dataset into 40 tasks of 5 classes following the setup from WSN (2022), La-MaML (2020) and FS-DGPM (2021), while FeCAM used 100 initial classes and prepared 5, 10 and 20 incremental learning settings with 20, 10 and 5 classes, respectively. FeCAM achieved in these scenarios 59.6%, 59.4% and 59.3%, respectively. We obtained 76.22% with HyperMask, but it is necessary to emphasize that a fair comparison of these approaches is not possible due to the above-mentioned issues. Similarly, the authors of FeCAM did not compare its approach with PackNet or WSN because these methods do not work with a pre-trained ViT.
> > > >
> > > > - We will also include FeCAM in our Related Works section.

---

### Official Review · Reviewer_uLSz · 2023-10-31

**Soundness:** 2 fair
**Presentation:** 1 poor
**Contribution:** 2 fair
**Rating:** 3
**Confidence:** 4

**Summary:**

This paper introduces a novel method, HyperMask, which leverages hypernetwork paradigms to model lottery ticket-based subnetworks.
HyperMask retains the ability to reuse weights from the lottery ticket module and adapt to new tasks, inheriting the strengths of both approaches. The semi-binary masks generated by HyperMask enhance the target network's ability to discriminate between classes in consecutive continual learning tasks. The paper offers a promising solution to the issue of catastrophic forgetting by combining two existing paradigms in a novel way, demonstrating potential significance in the field of continual learning.

**Strengths:**

1. The paper introduces an innovative approach, HyperMask, that combines the concepts of hypernetworks and the lottery ticket hypothesis. This unique combination leads to a novel method for addressing catastrophic forgetting in continual learning.
2. The idea of using semi-binary masks generated by a hypernetwork to create target subnetworks is a fresh and creative approach to tackling the challenges of continual learning.
3. The paper demonstrates a high level of quality in the experimental evaluation of HyperMask. It provides a comprehensive set of experiments on multiple benchmark datasets, comparing HyperMask against various state-of-the-art baseline methods.

**Weaknesses:**

1.The primary contribution of HyperMask, which involves using hypernetworks to produce semi-binary masks for continual learning, may not be considered highly novel in the field of continual learning and neural network architectures. Hypernetworks have been explored in prior research as a means to generate task-specific weights for neural networks [1][2], and the concept of using masks or pruning for model adaptation is not entirely new.
2. The paper lacks a deeper theoretical analysis of the proposed HyperMask method. It would be beneficial to include a more comprehensive theoretical foundation for the approach, explaining why semi-binary masks generated by hypernetworks are effective in minimizing forgetting.
3. The paper could benefit from providing more detailed guidelines for selecting hyperparameters. HyperMask involves parameters such as β, λ, and p, and while the authors mention the hyperparameter optimization process, they do not offer specific recommendations or insights on how to choose these hyperparameters effectively.
4. The paper could expand the comparison to include other hypernetwork variants or architectures that have been proposed in the literature. Specifically, discussing how HyperMask compares to variations of hypernetwork-based approaches.
5. The paper does not provide detailed information on the computational resources required for training HyperMask, including information on training time, memory usage.

[1] PackNet: Adding Multiple Tasks to a Single Network by Iterative Pruning (CVPR2018)
[2] Piggyback: Adapting a Single Network to Multiple Tasks by Learning to Mask Weights (ECCV2018)

**Questions:**

1. Can the authors provide a more detailed discussion of the novelty of their proposed method compared to prior work in the field of continual learning and hypernetwork-based approaches?
2. The paper mentions that HyperMask has some limitations in terms of memory consumption due to the requirement for the hypernetwork's output layer to match the number of parameters in the target network. Are there any potential strategies or approaches to mitigate this memory consumption issue?
3. The paper mentions that HyperMask has some limitations in terms of memory consumption due to the requirement for the hypernetwork's output layer to match the number of parameters in the target network. Are there any potential strategies or approaches to mitigate this memory consumption issue?

---

> ### Author Response · Authors · 2023-11-17
> **Response to Reviewer uLSz, part 1/2**
>
> We thank the reviewer for the feedback regarding our paper and constructive remarks.  We are glad that you appreciate our approach. We respond to your questions in the points below.
>
> *Q1*: Can the authors provide a more detailed discussion of the novelty of their proposed method compared to prior work in the field of continual learning and hypernetwork-based approaches?
>
> *R1*: The model combines two continual learning (CL) methodologies and provides a new concept using a meta-model to produce subnetworks. **The masks produced by the hypernetwork modulate the weights of the main network and act like dynamic filters, enhancing the target weights important for a given task and decreasing the importance of the remaining weights.**
>
> ### Architecture-based CL methods
> + To the best of our knowledge, our model is the first architecture-based CL model that uses hypernetwork, or, in general, any meta-model, for producing masks for other networks.
> + Hypernetwork is a meta-model that builds masks by sharing information across subsequent tasks in the meta-model’s weights. Therefore, knowledge from previous tasks is stored in the hypernetwork weights and the masks produced are not completely independent.
> + In consequence, we do not freeze any target model weights like in many architecture-based models.
>
> ### Hypernetwork-based methods
> + A single hypernetwork model produces sets of weights dedicated to each CL task. **Updates of hypernetworks are prepared not directly for the weights of the main model, like in HNET [1], but for masks modulating the target model.**
> + HyperMask, unlike most of the methods used, produces trainable updates for a trainable model without freezing in any part.
> + We show that hypernetworks may be used completely differently than classical hypernetwork CL models like HNET [1].
>
> [1] J. von Oswald et al., Continual learning with hypernetworks, ICLR 2020.
>
> *Q2*: The paper mentions that HyperMask has some limitations in terms of memory consumption due to the requirement for the hypernetwork's output layer to match the number of parameters in the target network. Are there any potential strategies or approaches to mitigate this memory consumption issue?
>
> *R2*:
> * We currently require that the hypernetwork output layer should match the number of parameters in the target network. Applying **chunked hypernetworks** similarly to [1] may mitigate this issue. In this architecture, the weights of the target model are created partially through so-called chunks; the number of chunks is another hyperparameter. Furthermore, other embedding vectors related to the chunk position must be created because the hypernetwork has to get information on which part of the target model weights will be produced. Similarly to the task embedding vectors, they are learned through back-propagation. Chunked hypernetworks will contribute to memory saving due to a lower number of hypernetwork weights (the smaller number of output neurons than in the case of the full hypernetwork architectures). Our initial experiments with chunked hypernetworks did not produce sufficient enough results in terms of classification accuracy but a more thorough analysis may solve this problem.
>
> * We need a similar calculation time as in HNET [1] and more detailed results for calculations performed on a single GPU card are presented in the following table:
>
> | Dataset              | Mean calculation time (with std dev.) in HH:MM:SS |
> |----------------------|---------------------------------------------------|
> | Split MNIST          | 00:21:06 +/- 00:02:37                             |
> | Permuted MNIST       | 01:45:14 +/- 00:04:07                             |
> | CIFAR 100 (ResNet)   | 10:26:37 +/- 00:10:05                             |
> | CIFAR 100 (ZenkeNet) | 06:02:25 +/- 00:01:54                             |

---

> > ### Author Response · Authors · 2023-11-20
> > **Response to Reviewer uLSz, part 2/2**
> >
> > *Q3*: The paper could benefit from providing more detailed guidelines for selecting hyperparameters. HyperMask involves parameters such as β, λ, and p, and while the authors mention the hyperparameter optimization process, they do not offer specific recommendations or insights on how to choose these hyperparameters effectively.
> >
> > *R3*:
> > * The hypernetwork should consist of one or, at most, a few hidden layers with a relatively small number of neurons, for instance, 2 hidden layers with 100 neurons per one layer, like in Permuted MNIST. For smaller datasets, like Split MNIST, the hypernetwork should be even narrower, e.g. consisting of 2 hidden layers with 25 neurons.
> > * We experimented with β and λ varying from about 0.0001 to 1. The larger the values of these hyperparameters, the stronger the effect of the regularization, leading from a specific moment to weak results on the current task, despite less network susceptibility to catastrophic forgetting. The exact values depend on the data complexity, the number of tasks and the target architecture (multilayer perceptron, convolutional neural network or other) used. For instance, stronger regularisation should be considered with the increasing number of tasks. Also, we observed that for more complex datasets and architectures based on convolutions, higher values of β and λ have to be set, e.g. about 0.01.
> > * We primarily experimented with the sparsity parameter p equal to 0, 30 or 70. The most common options were 0 or 30, corresponding to no pruning or pruning of 30% of weights from consecutive target network layers.

---

> > ### Comment · Reviewer_uLSz · 2023-11-22
> > **Response to authors**
> >
> > Firstly, I appreciate your efforts and prompt responses. However, there are a few concerns that need to be addressed:
> > 1. I respectfully disagree with the assertion that the proposed method can be considered as the first architecture-based CL model utilizing hypernetworks. There have been previous architecture-based CL models [1][2]. But you ignore my questions about discussion with these popular method.
> > [1] PackNet: Adding Multiple Tasks to a Single Network by Iterative Pruning (CVPR2018)
> > [2] Piggyback: Adapting a Single Network to Multiple Tasks by Learning to Mask Weights (ECCV2018)
> > 2. I fully support Reviewer 1qn2's suggestion regarding comparing our approach with FeCAM, which is the latest research in the field of continual learning.
> >
> > Consequently, I will lower my score.

---

> > > ### Author Response · Authors · 2023-11-22
> > > **Response to Reviewer uLSz**
> > >
> > > Dear Reviewer,
> > >
> > > Thank you for your time and consideration of our work.
> > >
> > > *1.*
> > > - We did not claim that HyperMask is the first architecture-based CL model in general, but that it is **the first architecture-based method utilizing hypernetworks**. Both [1] and [2] do not use hypernetworks in their pipelines. We inspired our method with WSN and HNET, while WSN is a newer method than PackNet and PiggyBack and WSN may even be considered as their improvement, which is visible in Figure 1 in [3].
> > > - In PackNet, a new subset of network weights is selected during training consecutive tasks. Also, all weights filtered for a given task must remain unchanged in the following tasks and cannot be pruned in the future. However, the number of available weights, which were not considered important for the previous tasks, is getting smaller according to the increasing number of tasks. Therefore, the space for modifying weights for some last tasks may be relatively low. In HyperMask, **we can create any number of embeddings for the hypernetwork**. Also, **the target network weights are filtered dynamically**, i.e. for the i-th task, a given weight may be considered as significant, while for the (i+1)-task, it can be irrelevant and again, for the (i+2)-task, it may be important. Furthermore, except for sparsifying the target network weights, they are scored continuously. They do not have to be just switched on or off. Therefore, this concept is different than in the case of PackNet.
> > > - PiggyBack creates a set of binary masks, while HyperMask may not only switch on or off some weights of the target network (due to the sparsity parameter) but, **for the all important target weights, create its continuous score between 0 and 1**. Also, the crucial thing is that **in PiggyBack the weights of the base network are frozen while HyperMask trains simultaneously the target network and the hypernetwork**. Therefore, in HyperMask, during learning of the selected task, the target network stores the knowledge from the previous tasks. The above facts are huge differences between PiggyBack and HyperMask.
> > >
> > > [3] H. Kang et al., Forget-free Continual Learning with Winning Subnetworks, ICML 2022.
> > >
> > > *2.*
> > > - The experimental setting of FeCAM is completely different than HyperMask because, in FeCAM, a backbone consists of a ViT encoder pre-trained on ImageNet-21K, while in HyperMask, we trained our models from scratch. Therefore, a fair comparison of these approaches is not possible. Despite this fact, in Split CIFAR-100, FeCAM achieved 85.7% of classification accuracy, while in the case of HyperMask, it was 77.34%. Taking into account that we used a convolutional neural network as a backbone (not a transformer) and that we did not apply a transfer learning approach, the difference does not seem to be major. We emphasize that the CL settings and the way of training are different.
> > >
> > > - According to our experiments in TinyImageNet, which we additionally computed, we divided the dataset into 40 tasks of 5 classes following the setup from WSN (2022), La-MaML (2020) and FS-DGPM (2021), while FeCAM used 100 initial classes and prepared 5, 10 and 20 incremental learning settings with 20, 10 and 5 classes, respectively. FeCAM achieved in these scenarios 59.6%, 59.4% and 59.3%, respectively. We obtained 76.22% with HyperMask, but it is necessary to emphasize that a fair comparison of these approaches is not possible due to the above-mentioned issues.

---

> ### Author Response · Authors · 2023-11-22
> **Response to Reviewer uLSz**
>
> Dear Reviewer,
>
> Thank you for your time and effort in reading our response! We hope our response has addressed your concerns. If you still feel unclear or worried, please let us know; we would be happy to clarify further and discuss any additional questions. If you feel your concerns have been addressed, please kindly consider if it is possible to update your score.
>
> Thank you!

---

### Official Review · Reviewer_G6Eq · 2023-11-01

**Soundness:** 2 fair
**Presentation:** 2 fair
**Contribution:** 2 fair
**Rating:** 5
**Confidence:** 5

**Summary:**

The paper works on continual learning with hypernetworks. The main idea is to generate masks to obtain a subnetwork for the new task. The experiments are conducted on Permuted MNIST, Split MNIST and Split CIFAR-100.

**Strengths:**

(1) The hypernetworks are used for continual learning in a different way of generating hypermasks for each task.
(2) It connects to lottery ticket theory with a single network for continual learning.
(3) The method is easy to follow in general.

**Weaknesses:**

(1) There are some works mentioned in the related work using masks as an extension of the whole network, it is unclear what benefits hypernetwork can bring.
(2) There are several common loss functions are used in the method, and it is unclear if the improvements are from the proposed hypernetworks or the additional regularizations. There is no ablation study on these components.
(3) The experimental evaluation is very limited. It only compares with other methods on very tiny benchmarks. The performance gain from Table 1 seems not very significant. And it is hard to know how much more computation and complexity the method needs.

**Questions:**

The superiority of the method is not clear compared to other existing methods and the evaluation section is weak.

---

> ### Author Response · Authors · 2023-11-17
> **Response to Reviewer G6Eq, part 1/2**
>
> We thank the reviewer for the remarks regarding our paper, which will improve its quality. We answer all of your questions as follows.
>
> *Q1*: There are some works mentioned in the related work using masks as an extension of the whole network, it is unclear what benefits hypernetwork can bring.
>
> *R1*: The model combines two continual learning (CL) methodologies and provides a new concept using a meta-model to produce subnetworks. **The masks produced by the hypernetwork modulate the weights of the main network and act like dynamic filters, enhancing the target weights important for a given task and decreasing the importance of the remaining weights.**
>
> ### Architecture-based CL methods
> + To the best of our knowledge, our model is the first architecture-based CL model that uses hypernetwork, or, in general, any meta-model, for producing masks for other networks.
> + Hypernetwork is a meta-model that builds masks by sharing information across subsequent tasks in the meta-model’s weights. Therefore, knowledge from previous tasks is stored in the hypernetwork weights and the masks produced are not completely independent.
> + In consequence, we do not freeze any target model weights like in many architecture-based models.
>
> ### Hypernetwork-based methods
> + A single hypernetwork model produces sets of weights dedicated to each CL task. **Updates of hypernetworks are prepared not directly for the weights of the main model, like in HNET [1], but for masks modulating the target model.**
> + HyperMask, unlike most of the methods used, produces trainable updates for a trainable model without freezing in any part.
> + We show that hypernetworks may be used completely differently than classical hypernetwork CL models like HNET [1].
>
> [1] J. von Oswald et al., Continual learning with hypernetworks, ICLR 2020.
>
> *Q2*: There are several common loss functions are used in the method, and it is unclear if the improvements are from the proposed hypernetworks or the additional regularizations. There is no ablation study on these components.
>
> *R2*:
> * Hypernetwork regularization using previously trained embeddings is a classical approach in hypernetwork models, while L1 regularization of the target model is an often used approach in CL. These two components are combined with the model and together form the presented method consisting of two trainable networks. **The first component is responsible for the regularization of the hypernetwork weights producing masks, while the second one accounts for the target network weights.** Therefore, both parts are necessary for HyperMask because it consists of two trainable networks and we have to prevent radical changes in their weights. In HNET [1], hypernetworks directly produce the target model’s weights and additional regularization is unnecessary because only one model is trained.
>
> * When we do not regularize the hypernetwork or main model, the weights are changed too drastically and catastrophic forgetting occurs. We present an ablation study for different hyperparameter settings responsible for the regularization strength (3 runs per one setup) on the Permuted MNIST dataset with 10 tasks using the same architecture, for which we presented results in Table 1 in our paper.
>
> | beta | lambda | Mean accuracy (with std dev.) |
> |------|--------|-------------------------------|
> | 0.01 | 0      | 84.18 +/- 0.18                |
> | 0.1  | 0      | 80.31 +/- 1.11                |
> | 1    | 0      | 73.68 +/- 0.31                |
> | 0    | 0.01   | 37.82 +/- 1.72                |
> | 0    | 0.1    | 46.32 +/- 1.07                |
> | 0    | 1      | 11.35 +/- 0.00                |

---

> ### Author Response · Authors · 2023-11-17
> **Response to Reviewer G6Eq, part 2/2**
>
> *Q3*: The experimental evaluation is very limited. It only compares with other methods on very tiny benchmarks. The performance gain from Table 1 seems not very significant. And it is hard to know how much more computation and complexity the method needs.
>
> *R3*:
> * We followed the evaluation protocols from HNET [1] and WSN [2]. Also, we performed experiments on TinyImageNet, a demanding dataset consisting of 40 continual learning tasks with 5 classes. A single image has a shape of 64 x 64 x 3. We prepared experiments using ResNet-20 architecture with an embedding vector of 96, β and λ equalling 1 and 0.1, respectively. Furthermore, p was set to 0, batch size to 16 and learning rate to 0.0001. Similar to WSN, the number of training epochs for each task was 10, and 50 validation samples from each class were selected in each CL task.
>
> * **We achieved state-of-the-art results, outperforming all reference methods.** The mean classification accuracy for 5 runs of HyperMask was equal to **76.22%**, while the best baseline, WSN, reached 71.96% using the architecture with 4 convolutional and 3 fully connected layers. Full results for **TinyImageNet** are depicted in the following table:
>
> ## Results for TinyImageNet
> | Method        | Mean classification accuracy in % (with std dev.) |
> |---------------|---------------------------------------------------|
> | La-MaML [3]   | 66.90 +/- 1.65                                    |
> | GPM           | 67.39 +/- 0.47                                    |
> | FS-DGPM [4]   | 70.41 +/- 1.30                                    |
> | PackNet       | 55.46 +/- 1.22                                    |
> | SupSup        | 59.60 +/- 1.05                                    |
> | WSN, c = 3%   | 68.72 +/- 1.63                                    |
> | WSN, c = 5%   | 71.22 +/- 0.94                                    |
> | WSN, c = 10%  | 71.96 +/- 1.41                                    |
> | WSN, c = 30%  | 70.92 +/- 1.37                                    |
> | WSN, c = 50%  | 69.06 +/- 0.82                                    |
> | **HyperMask** | **76.22 +/- 1.06**                                |
>
> * In addition to the results for Split CIFAR-100 and Tiny ImageNet, we show in the paper results for Permuted MNIST in 100 tasks with 10 classes and they are presented in Figure 2. **It confirms that our model may be trained on continual learning datasets with many tasks.** Especially interesting is the high result for the first task despite learning 99 of the subsequent ones. We additionally analyze this phenomenon with the t-SNE method for Permuted MNIST with 10 tasks (Figure 3) and Split MNIST with 5 tasks (Figure 6).
>
> * The classical continual learning benchmarks are quite old and we think that state-of-the-art CL models are close to optimal results in such tasks. Our results are strong because we have two best results and two the second ones. We also achieved competitive results in a larger benchmark, i.e. TinyImageNet.
>
> * We need a similar calculation time as in HNET [1] and more detailed results for calculations performed on a single GPU card are presented in the following table:
>
> | Dataset              | Mean calculation time (with std dev.) in HH:MM:SS |
> |----------------------|---------------------------------------------------|
> | Split MNIST          | 00:21:06 +/- 00:02:37                             |
> | Permuted MNIST       | 01:45:14 +/- 00:04:07                             |
> | CIFAR 100 (ResNet)   | 10:26:37 +/- 00:10:05                             |
> | CIFAR 100 (ZenkeNet) | 06:02:25 +/- 00:01:54                             |
>
> * According to the memory consumption, currently, we require that the hypernetwork output layer should match the number of parameters in the target network. Applying **chunked hypernetworks** similarly to [1] may mitigate this issue. In this architecture, the weights of the target model are created partially through so-called chunks; the number of chunks is another hyperparameter. Furthermore, other embedding vectors related to the chunk position must be created because the hypernetwork has to get information on which part of the target model weights will be produced. Similarly to the task embedding vectors, they are learned through back-propagation. Chunked hypernetworks will contribute to memory saving due to a lower number of hypernetwork weights (the smaller number of output neurons than in the case of the full hypernetwork architectures). Our initial experiments with chunked hypernetworks did not produce sufficient enough results in terms of classification accuracy but a more thorough analysis may solve this problem.
>
> [2] H. Kang et al., Forget-free Continual Learning with Winning Subnetworks, ICML 2022.
>
> [3] G. Gupta et al., La-MaML: Looking ahead meta learning for continual learning, In Advances in NeurIPS, 2020.
>
> [4] D. Deng et al., Flattening sharpness for dynamic gradient projection memory benefits continual learning, In Advances in NeurIPS, 2021.

---

> > ### Comment · Reviewer_G6Eq · 2023-11-22
> >
> > Thanks for your efforts in providing more detailed results and explanations.

---

> ### Author Response · Authors · 2023-11-22
> **Response to Reviewer G6Eq**
>
> Dear Reviewer,
>
> Thank you for your time and effort in reading our response! We hope our response has addressed your concerns. If you still feel unclear or worried, please let us know; we would be happy to clarify further and discuss any additional questions. If you feel your concerns have been addressed, please kindly consider if it is possible to update your score.
>
> Thank you!

---

### Author Response · Authors · 2023-11-17
**The response to all Reviewers**

We thank all the reviewers for their feedback and comments on our paper.

According to the suggestions of Reviewers G6Eq and 1qn2, **we performed additional experiments with a more challenging dataset, TinyImageNet**, that was divided into 40 CL tasks with 5 classes. We achieved a **very high mean classification accuracy, outperforming all reference methods**. The presented experiments confirm that HyperMask is a high-performing continual learning method.

We added an explanation of the novelty of HyperMask. We emphasized the differences between other approaches and highlighted new features of our algorithm.

We also calculated the mean model training time and explained a method of reducing memory consumption with chunked hypernetworks.

We addressed the individual remarks of the reviewers, such as the ablation study regarding loss function components, recommendations about hyperparameter selection and others.

We will be glad if the reviewers get acquainted with the corrections and newly presented results.

---

### Meta-Review · Area_Chair_mXXp · 2023-12-10

**Metareview:**

The paper tackles continual learning with hypernetworks. The idea is to use the same network to generate semi-binary masks to obtain subnetworks for new tasks in continual learning. The semi-binary masks generated by HyperMask enhance the target network's ability to discriminate between classes in consecutive continual learning tasks. The paper offers a promising solution to the issue of catastrophic forgetting that often plagues continual learning. By combining two existing ideas in a new way, the paper demonstrates potential significance in the field of continual learning.

In terms of results, the experiments are conducted on Permuted MNIST, Split MNIST, Split CIFAR-100, and TinyImageNet. The result does not seem to be strong enough, and is lacking important benchmarks.

That said, we do think the paper post rebuttal is much stronger. We encourage the authors to improve the method, enhance the experiments, and look forward to seeing a future submission of this method.

**Justification For Why Not Higher Score:**

While it is a reasonable idea to leverage HyperNetworks to generate masks for continual learning, both ideas (using hypernetworks for CL as well as generating masks for CL) are not new, and the proposed method does not sufficiently differentiate itself with examples like HNET and SupSup.

The writing and presentation can also be improved. The citation format for example is wrong throughout. The literature review section should also highlight the difference between existing methods and the proposed method.

**Justification For Why Not Lower Score:**

N/A

---

### Decision · Program_Chairs · 2024-01-16

Reject